# Beyond 'Templates': Category-Agnostic Object Pose, Size, and Shape Estimation from a Single View

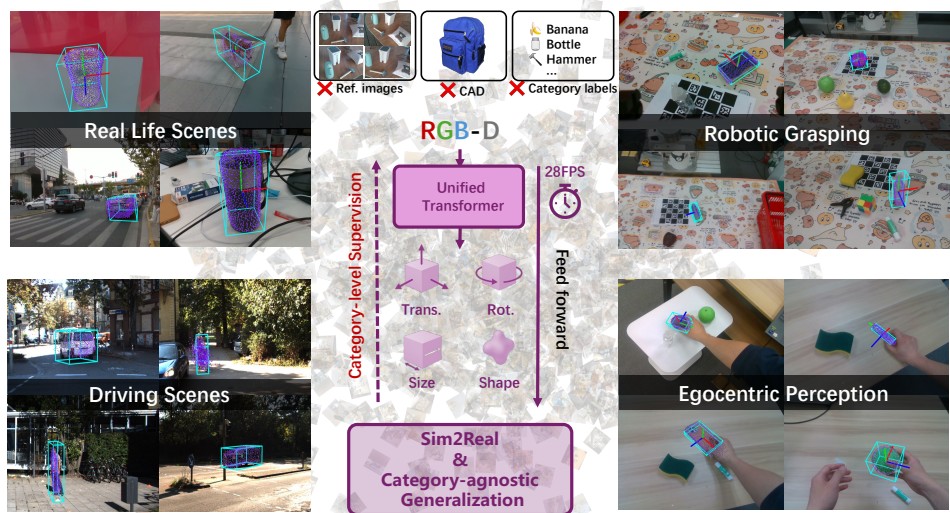

Figure 1: Results on diverse domain datasets using our end-to-end regression-based framework. Trained exclusively on a large synthetic dataset, our model generalizes effectively to unseen object categories across multiple real-world domains, including daily-life scenes, autonomous driving, robotic manipulation, and egocentric video data.

## Abstract

Estimating an object's 6D pose, size, and shape from visual input is a fundamental problem in computer vision, with critical applications in robotic grasping and manipulation. Existing methods either rely on object-specific priors such as CAD models or templates, or suffer from limited generalization across categories due to pose–shape entanglement and multi-stage pipelines. In this work, we propose a **unified, category-agnostic framework** that simultaneously predicts 6D pose, size, and dense shape from a single RGB-D image, without requiring templates, CAD models, or category labels at test time. Our model fuses dense 2D features from vision foundation models with partial 3D point clouds using a Transformer encoder enhanced by a Mixture-of-Experts, and employs parallel decoders for pose–size estimation and shape reconstruction, achieving **real-time inference at 28 FPS**. Trained solely on synthetic data from 149 categories in the SOPE dataset, our framework is evaluated on five diverse benchmarks SOPE, ROPE, Objaverse-Pose, HANDAL and HouseCat6D, spanning 300+ categories. It achieves **state-of-the-art accuracy on seen categories** while demonstrating **remarkably strong zero-shot generalization** to unseen real-world objects, establishing a new standard for open-set 6D understanding in robotics and embodied AI.

## 1 Introduction

Estimating the pose, size, and shape of objects from visual input is a fundamental challenge in computer vision, underpinning robotic grasping Cheang et al. (2022); Sun et al. (2023); Zhang et al. (2023; 2024a); Irshad et al. (2022a) and manipulation Lin et al. (2023; 2022a); Wang et al. (2024); Wen et al. (2023); Huang et al. (2025b;a), as shown in Fig. 1. Yet despite decades of progress, current

systems remain limited in their ability to scale beyond carefully curated settings. Instance-level 6D pose methods often rely on reference images, templates, or object-specific CAD models Labbé et al. (2022); Wen et al. (2024); Nguyen et al. (2022), which provide strong priors but are rarely available in open-set, and real-world environments. Category-level approaches relax these constraints by leveraging canonical supervision across classes Wang et al. (2019b); Jung et al. (2024), but they inherit two persistent bottlenecks: (i) pose–shape entanglement due to large intra-class variations and partial observations, and (ii) dependence on multi-stage Labbé et al. (2022); Wen et al. (2024); Nguyen et al. (2022); Lee et al. (2025); Wen et al. (2023) or diffusion-based pipelines Zhang et al. (2023; 2025a), which restrict efficiency and real-time deployment.

This gap highlights a central open question: *Can we unify pose, size, and shape estimation into a single, real-time framework that generalizes to unseen categories without requiring test-time priors?*

To address these limitations, we present a more scalable and practical approach to category-level pose and size estimation. Our model is trained using category-level canonical supervision, but is able to perform category-agnostic inference from a single RGB-D image—without requiring CAD models, reference views, or category labels at test time. This design preserves category-level consistency during training, while its category-agnostic inference enables generalization to previously unseen categories, facilitating open-set 6D understanding without reference priors.

In this work, we answer this question affirmatively. We introduce a scalable, category-agnostic framework that infers an object's full 6D pose, size, and shape from a single RGB-D image without templates, CAD models, or category labels at test time. Our design marries dense 2D features from vision foundation models with partial 3D point clouds, processed through a Transformer encoder augmented by a Mixture-of-Experts (MoE) for scalable specialization across diverse shape distributions. Two lightweight decoders jointly predict the 6D pose–size estimate and reconstruct object shape, achieving unified reasoning in a single forward pass at 28 FPS. This simplicity contrasts sharply with cascaded or iterative pipelines Labbé et al. (2022); Wen et al. (2024); Nguyen et al. (2022); Lee et al. (2025); Wen et al. (2023); Zhang et al. (2023; 2025a), making the approach both robust and practical.

Trained purely on synthetic data from 149 SOPE categories Zhang et al. (2025b), our model is evaluated across five diverse benchmarks: SOPE Zhang et al. (2025b), ROPE Zhang et al. (2025b), ObjaversePose, HANDAL Guo et al. (2023) and HouseCat6D Jung et al. (2024), spanning 300+ categories and synthetic-to-real transfer. It not only achieves state-of-the-art accuracy on seen categories, but also demonstrates remarkably strong zero-shot generalization to novel real-world objects, substantially outperforming prior category-level methods as well as reference-based novel object pose estimators as in Fig. 3. These results position our framework as a decisive step toward open-set 6D understanding: real-time, category-agnostic perception that is both scalable and robust in the complexity of the world.

**Contributions.** Our contributions are fourfold. First, we propose the first **unified, category-agnostic framework** that simultaneously estimates an object's 6D pose, size, and shape from a single RGB-D image—without requiring CAD models, templates, or category labels at test time. Second, we design a **scalable architecture** that fuses 2D foundation-model features with 3D point clouds via a Transformer encoder enhanced by a Mixture-of-Experts, enabling efficient specialization across diverse shape distributions and **real-time inference at 28 FPS** through a single forward pass. Third, we demonstrate **extensive generalization and state-of-the-art performance**: trained exclusively on synthetic SOPE data, our model achieves leading accuracy on SOPE, ROPE, ObjaversePose, HANDAL benchmarks spanning 300+ categories, while delivering **remarkably strong zero-shot transfer** to unseen real-world objects. Finally, we introduce **ObjaversePose**, a synthetic dataset built from Objaverse CAD models under the SOPE canonical convention, rendering photorealistic RGB-D from 20 views per object to provide **greater geometric and semantic diversity** for category-agnostic 6D estimation.

## 2 RELATED WORK

**Category-Level 6D Pose Estimation.** Category-level methods aim to generalize pose estimation across unseen object instances within a category Wang et al. (2019b; 2022); Jung et al. (2024). Early works such as NOCS Wang et al. (2019b) introduced the notion of canonical space to en-

able pose alignment without requiring CAD models. Follow-up approaches Lin et al. (2022a); Liu et al. (2023); Chen et al. (2021); Zhang et al. (2023) leveraged point cloud geometry and symmetry-aware losses to improve generalization. Some methods further incorporate shape reasoning Tian et al. (2020); Irshad et al. (2022b) to jointly predict object size and shape. While these methods remove the need for object-specific models, most are limited to seen categories and are trained on relatively small sets of object types, restricting generalization to novel classes. Recent works have attempted to scale category-level 6D learning using large-scale datasets Zhang et al. (2025b); Krishnan et al. (2024), but inference-time generalization remains a challenge. Diffusion-based methods such as GenPose Zhang et al. (2023) and GenPose++ Zhang et al. (2025a) model pose and size as a multi-modal distribution, but require iterative sampling, auxiliary scoring networks, and multi-stage training. In contrast, our method offers a unified, one-pass framework that regresses pose, size, and shape in real time, while generalizing to unseen object categories under a category-agnostic inference setting.

**Instance-Level Novel Object Pose Estimation.** Another line of work targets zero-shot pose estimation for novel instances using CAD models Labbé et al. (2022); Nguyen et al. (2024), single or multi-view references Lee et al. (2025); Liu et al. (2025); He et al. (2022a). These approaches often reconstruct object geometry—either explicitly via 3D modeling Liu et al. (2022); Li et al. (2023) or implicitly through image-based retrieval He et al. (2022b); Nguyen et al. (2022). FoundationPose Wen et al. (2024) supports either CAD or reference images, combining model-based and model-free paradigms. However, such methods generally require additional inputs at inference time and rely on alignment with specific object instances. Unlike these methods, we do not assume access to any object-specific references. Our model is trained entirely on synthetic data using category-level canonical supervision, and performs category-agnostic inference from a single RGB-D image—without requiring CAD models, reference views, or category information at test time. This makes our method more deployable in open-set, real-world scenarios.

# 3 METHOD

**Architecture Overview**. Given an RGB image patch $\mathbf{I} \in \mathbb{R}^{H \times W \times 3}$ and a partially observed point cloud $\mathbf{P} \in \mathbb{R}^{N \times 3}$, our goal is to simultaneously estimate the object's 6D pose $\{\mathbf{R}, \mathbf{t}\} \in \mathrm{SE}(3)$, its 3D size $\mathbf{s} \in \mathbb{R}^3$, and the complete shape $\mathbf{P}_{\mathrm{dense}} \in \mathbb{R}^{N_d \times 3}$ of the object in the camera frame. Here, $\mathbf{R} \in \mathrm{SO}(3)$ represents the 3D rotation, while $\mathbf{t} \in \mathbb{R}^3$ denotes the 3D translation. The groups $\mathrm{SE}(3)$ and $\mathrm{SO}(3)$ refer to 3D rigid transformations and 3D rotations, respectively.

**Motivation.** Given a partial point cloud and a single-view RGB image, we obtain limited surface information about an object's shape. In practical robotic manipulation, recovering the complete object shape is crucial for generating accurate grasp poses, particularly for multi-fingered dexterous hands. Motivated by this, our framework jointly infers an object's 6D pose, size, and full shape from partial observations. This process mirrors human perception: even from a single viewpoint, humans can mentally reconstruct an object's complete geometry by leveraging visual cues and prior knowledge of familiar shapes encountered in daily life. We detail our method in the following sections.

## 3.1 FEATURE EXTRACTION

Our goal is to learn a category-agnostic representation that enables universal estimation of object shape, pose, and size. So we leverage the foundation model RADIOv2.5 Heinrich et al. (2025), which extracts generalizable and category-agnostic local features as the prior. RADIOv2.5 distills knowledge from several powerful 2D vision models Ravi et al. (2024); Oquab et al. (2023); Radford et al. (2021); Fang et al. (2023), combining the dense feature extraction capability of SAM Ravi et al. (2024) with the SE(3)-consistent semantic features of DINOv2 Oquab et al. (2023). As shown in Zhang et al. (2024b;c), DINOv2 captures SE(3)-consistent local features that are particularly useful for establishing semantic correspondences across objects of varying shapes and poses, which aligns well with the SE(3)-invariant nature of NOCS coordinates. Given an RGB image patch $\mathbf{I} \in \mathbb{R}^{H \times W \times 3}$, we use the RADIOv2.5 encoder $\mathbf{E}_{\mathrm{RADIO}}(\cdot)$ to extract semantic feature maps $\mathbf{F}_{\mathrm{rgb}} \in \mathbb{R}^{h \times w \times 1024}$:

$$\mathbf{F}_{\mathrm{rgb}} = \mathrm{Concate}(\mathbf{E}_{\mathrm{RADIO}}(\mathbf{I})_i), \tag{1}$$

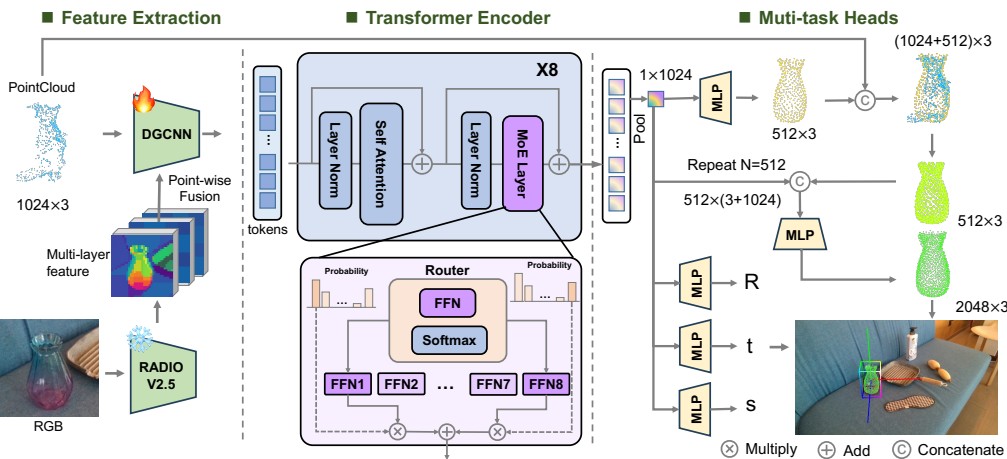

Figure 2: **Framework Overview.** Given a cropped RGB image and its corresponding segmented point cloud, the model first extracts dense 2D features using RADIOv2.5 Heinrich et al. (2025), which are concatenated with 3D point coordinates. A DGCNN processes the fused input to produce keypoint-aware features, forming object tokens. These tokens are passed through a Transformer encoder with a Mixture-of-Experts (MoE) module to produce a global object representation. Two parallel decoder branches predict (i) the 6D pose and size via direct regression, and (ii) the object shape in two stages: a coarse shape prediction followed by refinement using fused points. The entire pipeline is fully end-to-end and operates in real time.

where $i \in \{8, 16, 23\}$ denotes the transformer layers used for feature extraction, following Heinrich et al. (2025). Outputs from these layers are concatenated to enrich the semantic features.

Next, we fuse these RGB features with the corresponding input point cloud in a point-wise manner, following DenseFusion Wang et al. (2019a). The resulting feature-enriched points are passed through a Dynamic Graph Convolutional Neural Network (DGCNN) encoder $\mathbf{E}_{\text{GCN}}(\cdot)$:

$$\mathbf{F}_{\text{fuse}} = \mathbf{E}_{\text{GCN}}(\{\mathbf{P}_i \odot \mathbf{F}_{\text{rgb}}\}), \quad i = 1, \dots, N, \tag{2}$$

where $\odot$ denotes concatenation. The fused embedding $\mathbf{F}_{\text{fuse}} \in \mathbb{R}^{n \times d}$, where $n = 128$ is the number of tokens and $d = 256$ is the embedding dimension, serves as the input tokens for the transformer encoder.

## 3.2 TRANSFORMER ENCODING WITH MOE

Given a point cloud enriched with semantic RGB features, we first extract fused input embeddings using a DGCNN backbone, following Yu et al. (2021). These embeddings are subsequently processed by a stack of Transformer blocks with multi-head self-attention Vaswani et al. (2017), in the spirit of Yu et al. (2021; 2023).

To enhance modeling capacity while maintaining computational efficiency, we replace the standard feed-forward layers in each Transformer block with Mixture-of-Experts (MoE) layers, inspired by the success of MoE architectures in large language models Du et al. (2022); Fedus et al. (2022); Liu et al. (2024). Each MoE layer contains an independent set of experts and uses a linear gating network to compute softmax-normalized logits over $n = 8$ experts. For each token, we select the top-2 experts according to the gating scores, route the token to these experts, and aggregate their outputs weighted by the normalized gating values. Following standard MoE designs Shazeer et al. (2017), we add noise to the gating function to encourage better load balancing across experts. This design enables the model to specialize across diverse object types and shape patterns, improving performance with minimal overhead. Finally, the output features are aggregated via global pooling to produce a compact global representation for downstream 6D pose, size, and shape estimation.

## 3.3 MULTI-TASK HEADS

We leverage a large-scale, diverse dataset covering hundreds of object categories and shapes, allowing our model to implicitly acquire a rich shape prior across varied shape distributions. To jointly capture object pose, size, and shape, we employ a **multi-task decoding head** that regresses all three quantities directly from the global feature representation, enabling unified reasoning in a single forward pass.

**Shape Reconstruction.** From the extracted global feature vector, we first regress a coarse complete object shape $\mathbf{P}_{\text{coarse}} \in \mathbb{R}^{512 \times 3}$ using a lightweight MLP. Recognizing that the input partial point cloud $\mathbf{P}$ contains complementary geometric cues, we concatenate $\mathbf{P}_{\text{coarse}}$ with $\mathbf{P}$ to form a combined point set, which is then processed through another MLP with a Sigmoid activation to produce a confidence score for each point. This allows the network to select the most reliable points for fusion, producing the refined fused point cloud $\mathbf{P}_{\text{fuse}} \in \mathbb{R}^{512 \times 3}$. Finally, each point in $\mathbf{P}_{\text{fuse}}$ is augmented with the global feature vector and passed through a final MLP to regress the dense, high-resolution point cloud $\mathbf{P}_{\text{dense}} \in \mathbb{R}^{2048 \times 3}$, representing the predicted complete object shape. This confidence-guided fusion mechanism effectively integrates partial observations with learned priors, enabling robust shape reconstruction even under severe occlusion.

**Pose and Size Estimation.** Supervised by the shape reconstruction head, the transformer encoder learns global features that encode comprehensive shape, pose, and size information from a single camera view. To explicitly predict object pose and size, we introduce a dedicated decoding branch that regresses rotation, translation, and scale directly from the global feature vector. For rotation, we adopt the continuous 6D representation Zhou et al. (2019), which uses the first two columns of the rotation matrix $\mathbf{R} \in \mathrm{SO}(3)$ and reconstructs a valid matrix via orthogonalization, improving training stability and prediction accuracy. By jointly learning shape, pose, and size in a unified framework, our model captures inter-dependencies between geometry and spatial configuration, enhancing both robustness and generalization to unseen categories.

## 3.4 LOSS FUNCTIONS

**Reconstruction Loss.** For the point cloud reconstruction task, we adopt the Chamfer Distance with L1 norm, following the approach in Yu et al. (2021), to supervise both the coarse and dense point cloud outputs. Specifically, we define two separate reconstruction losses: one for the coarse predicted point cloud $\mathbf{P}_{\text{coarse}}$ and another for the final dense reconstruction $\mathbf{P}_{\text{dense}}$. The Chamfer Distance measures the average closest-point distance between the predicted and ground truth point sets, encouraging accurate shape recovery at different stages of the pipeline.

The reconstruction losses are formulated as follows:

$$\mathcal{L}_{\text{recon1}} = \text{Chamfer}(\mathbf{P}_{\text{coarse}}, \mathbf{P}_{\text{coarse}}^{\text{gt}}), \tag{3}$$

$$\mathcal{L}_{\text{recon2}} = \text{Chamfer}(\mathbf{P}_{\text{dense}}, \mathbf{P}_{\text{dense}}^{\text{gt}}). \tag{4}$$

Here, $\mathbf{P}_{\{\cdot\}}^{\text{gt}}$ denotes the ground truth complete point cloud. These losses jointly guide the model to generate increasingly accurate shapes from coarse to fine resolution.

**Pose and Size Regression Loss.** For the pose estimation component, we use the Smooth L1 loss instead of the standard L2 loss, as our empirical results show that L2 loss leads to suboptimal performance in this task. In addition, when computing the loss for predicted rotation matrices, we account for object symmetries to reduce ambiguity, following the strategy in Zhang et al. (2025b). Specifically, for each of the axes, we categorize an object's symmetry as one of the following: no symmetry, 90-degree rotational symmetry, 180-degree rotational symmetry, or arbitrary-angle rotational symmetry. Based on this classification, we generate a set of valid ground truth rotation matrices and compare the predicted rotation against all candidates. The loss is computed with respect to the closest ground truth rotation. The pose and size estimation losses are defined as:

$$\mathcal{L}_{\text{rot}} = \min_{\mathbf{R}_i^{\text{gt}} \in \mathcal{G}_{\mathbf{R}}} \text{SmoothL1}(\mathbf{R}_{1,2}, (\mathbf{R}_i^{\text{gt}})_{1,2}), \tag{5}$$

$$\mathcal{L}_{\text{trans}} = \text{SmoothL1}(\mathbf{t}, \mathbf{t}^{\text{gt}}), \tag{6}$$

$$\mathcal{L}_{\text{size}} = \text{SmoothL1}(\mathbf{s}, \mathbf{s}^{\text{gt}}). \tag{7}$$

where $\mathbf{R}, \mathbf{t}, \mathbf{s}$ denote the predicted rotation matrix, translation, and size respectively, $\mathbf{t}^{\text{gt}}, \mathbf{s}^{\text{gt}}$ are the corresponding ground truth values. $\mathbf{R}_{1,2}$ denotes the first two columns of $\mathbf{R}$. $\mathcal{G}_{\mathbf{R}} = \{\mathbf{R}_i^{\text{gt}} \mid i \in \{1, 2, \ldots, M\}\}$ is the set of symmetric-equivalent ground truth rotation matrices, where $M$ denotes the total number of valid rotations derived from the object's symmetry type. The predicted rotation is compared against each candidate in $\mathcal{G}_{\mathbf{R}}$, and the loss is computed using the closest match.

**All Losses.** Our final objective function integrates losses from both point cloud reconstruction and 6D pose and size estimation. It is defined as the sum of the individual loss terms:

$$L = \mathcal{L}_{\text{recon1}} + \mathcal{L}_{\text{recon2}} + \mathcal{L}_{\text{rot}} + \mathcal{L}_{\text{trans}} + \mathcal{L}_{\text{size}}. \tag{8}$$

In practice, we found that simply setting all loss coefficients to 1 yields stable training and strong performance, without requiring additional balancing. By jointly optimizing these components, the model learns to reconstruct accurate 3D shapes while simultaneously estimating precise object poses and sizes. This unified objective enhances both the accuracy and robustness of the overall system.

## 4 EXPERIMENTS

### 4.1 EXPERIMENT SETTINGS

**Dataset**. We evaluate our method on five benchmarks: SOPE, ROPE, ObjaversePose, HANDAL and HouseCat6D. SOPE is a large-scale synthetic dataset with 5,000 instances across 149 categories and simulated depth. ROPE provides real-world scans of 580 objects with dense 6D pose annotations across diverse materials and backgrounds. ObjaversePose is introduced as a challenging benchmark for category-agnostic 6D pose perception on unseen object categories. Leveraging the diversity of Objaverse Deitke et al. (2022), it comprises 2,354 instances from 154 categories that extend beyond SOPE's taxonomy, offering substantial broader variation in shape, appearance, and semantics. Additional details are provided in Appendix 7.3. For evaluating zero-shot real-world generalization, we use the HANDAL dynamic onboarding subset, which contains novel categories absent from SOPE with high-quality RGB-D scans and pose/size annotations, serving as a challenging test for category-agnostic 6D estimation. We further demonstrate the differences between ObjaversePose/HANDAL and SOPE through semantic and geometric t-SNE comparisons in Appendix 7.4). To further evaluate real-world reconstruction, we test on HouseCat6D, which contains 194 household objects across 10 categories with high-quality CAD models.

**Evaluation Metric**. We evaluate our method using complementary metrics that jointly assess pose accuracy, size alignment, and shape reconstruction. For pose–size alignment, we report the Area Under the Curve (AUC) of 3D bounding box IoU Zhang et al. (2025a) at thresholds of 25, 50, and 75, which directly measures the consistency between predicted and ground-truth transformations. As a geometry-based criterion, 3D IoU offers a reliable and category-agnostic measure across both seen and unseen settings. To capture both rotational and translational precision, we adopt the Volume Under Surface (VUS) metric Zhang et al. (2025a) and report VUS@5°2cm, VUS@5°5cm, VUS@10°2cm, and VUS@10°5cm, quantifying the proportion of predictions within joint error thresholds; we further report mean rotation and translation errors over all test instances to complement these success rates. Finally, we evaluate reconstruction quality using the L1 Chamfer Distance (CDL1) Yu et al. (2021), which measures the geometric fidelity of predicted shapes.

**Experimental Setup**. All baselines, including our methods, are trained solely on the SOPE training split, with no additional finetuning on external datasets. FoundationPose and Any6D are the only exceptions, evaluated directly from their released checkpoints. Evaluation spans four test sets: SOPE (synthetic, seen categories), ROPE (real-world, seen), ObjaversePose (synthetic, unseen), and HANDAL dynamic onboarding (real-world, novel), covering instance- and category-level generalization across synthetic and real domains. We adopt complementary metrics: on

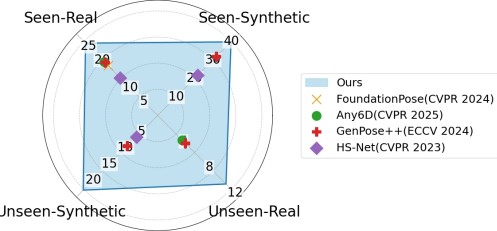

Figure 3: Radar chart comparing our method with baselines across four evaluation settings.

Table 1: Quantitative comparison of category-level pose estimation on the ROPE and SOPE datasets (seen categories, unseen instances). '-' indicates: (i) GenPose does not predict object scale; (ii) NOCS metrics are omitted due to large errors. $^*$ indicates our model with the shape head removed for fair FPS measurement.

| Dataset | Method | AUC ↑ | | | VUS ↑ | | | | Mean error ↓ | | FPS ↑ |
|---|---|---|---|---|---|---|---|---|---|---|---|
| | | IoU25 | IoU50 | IoU75 | 5°2cm | 5°5cm | 10°2cm | 10°5cm | rot(°) | trans(cm) | |
| ROPE | NOCS Wang et al. (2019b) | 0.0 | 0.0 | 0.0 | 0.0 | 0.0 | 0.0 | 0.0 | - | - | 5 |
| | SGPA Chen & Dou (2021) | 10.5 | 2.0 | 0.0 | 4.3 | 6.7 | 9.3 | 15.0 | 60.2 | 2.81 | 6 |
| | IST-Net Liu et al. (2023) | 28.7 | 10.6 | 0.5 | 2.0 | 3.4 | 5.3 | 8.8 | 78.4 | 3.40 | 35 |
| | HS-Pose Zheng et al. (2023) | 31.6 | 13.6 | 1.1 | 3.5 | 5.3 | 8.4 | 12.7 | 63.3 | 3.02 | **50** |
| | GenPose Zhang et al. (2023) | – | – | – | 6.6 | 9.6 | 13.1 | 19.3 | 42.2 | 2.05 | 6 |
| | GenPose++ Zhang et al. (2025a) | 39.0 | 19.1 | 2.0 | 10.0 | **15.1** | 19.5 | **29.4** | 30.7 | 1.28 | 4 |
| | **Ours** | **44.9** | **26.1** | **4.9** | **10.1** | 14.4 | **20.0** | 29.1 | **27.7** | **1.25** | 28$^*$ |
| SOPE | NOCS Wang et al. (2019b) | 0.0 | 0.0 | 0.0 | 0.0 | 0.0 | 0.0 | 0.0 | - | - | 5 |
| | SGPA Chen & Dou (2021) | 13.3 | 3.2 | 0.0 | 7.7 | 10.1 | 15.0 | 20.4 | 33.8 | 2.09 | 6 |
| | IST-Net Liu et al. (2023) | 36.5 | 16.9 | 1.4 | 3.6 | 5.1 | 8.6 | 11.4 | 60.4 | 3.72 | 35 |
| | HS-Pose Zheng et al. (2023) | 40.1 | 21.7 | 3.2 | 6.3 | 8.0 | 13.6 | 17.3 | 39.9 | 2.46 | **50** |
| | GenPose Zhang et al. (2023) | – | – | – | 11.9 | 14.4 | 21.2 | 26.3 | 26.1 | 1.62 | 6 |
| | GenPose++ Zhang et al. (2025a) | 50.1 | 31.9 | 6.4 | **18.4** | **23.0** | 31.9 | **40.2** | 19.9 | 1.14 | 4 |
| | **Ours** | **56.4** | **39.8** | **12.7** | **18.4** | 22.1 | **32.8** | 40.1 | **16.0** | **0.99** | 28$^*$ |

SOPE and ObjaversePose, we report AUC of 3D IoU, VUS, and mean rotation/translation errors; on ROPE and HANDAL, we report AUC of 3D IoU at thresholds 25/50/75 for comparison with reference-based methods. To assess robustness under occlusion, ObjaversePose is evaluated at varying visibility levels, simulating realistic single-view challenges.

## 4.2 RESULTS

**Category-level results on SOPE and ROPE.** We evaluate on synthetic SOPE and real-world ROPE to test generalization within seen categories across domains (Table 1). On SOPE, which matches the training distribution, our model surpasses GenPose++ across all metrics, achieving higher AUC, lower rotation/translation errors, and slightly better VUS. On ROPE, which contains real scans of novel instances, our method generalizes well despite the domain gap, again outperforming GenPose++ in AUC and mean errors, and showing strong shape completion on depth-missing objects from reflective/transparent surfaces 6. Although GenPose++ is stronger on some VUS thresholds (e.g., 5°5 cm), our model excels on others (e.g., 5°2 cm, 10°5 cm), yielding competitive overall accuracy. We attribute these gains to two design choices: (1) integrating DGCNN-based local geometry encoding with Transformer-based global context aggregation, which capture both fine-grained geometry and global context, and (2) a unified end-to-end pipeline that predicts pose, size, and shape simultaneously, avoiding error accumulation across stages. Compared to the multi-network design of GenPose++, our approach is simpler, faster (28 FPS vs. 4 FPS; measured in pose-only mode with all pose modules active and the completion head disabled), and more accurate. All FPS values are measured end-to-end with batch size 1, averaged over 1000 iterations after a 50-step warm-up with GPU synchronization.

**Category-agnostic generalization on ObjaversePose.** Table 2 reports AUC of 3D bounding box IoU on ObjaversePose under varying occlusion, evaluating generalization to unseen categories and robustness to partial observations. Our method consistently surpasses all baselines across all occlusion levels. At 0% occlusion, it achieves 42.2 AUC@IoU25, nearly doubling the best baseline GenPose++ (21.3). Even under 75% occlusion, it retains a clear margin. The gap further widens at stricter thresholds, reflecting stronger pose–size consistency. We attribute these gains to combining dense foundation features with geometry-aware point tokens and Transformer reasoning, which capture both global semantics and local shape cues.

**Comparison with Reference-Based Novel Object Pose Estimation Methods.** We compare our method against three state-of-the-art approaches for novel object pose estimation: Any6D Lee et al. (2025), a model-free method designed chiefly for single-reference inference, with auxiliary support for reference-free inference, and FoundationPose Wen et al. (2024), a reference-based approach that takes either CAD models or reference images, and GenPose++ Zhang et al. (2025a), the strongest category-agnostic baseline prior to our work. We evaluate under two conditions: (1)

Table 2: ObjaversePose (unseen categories) under varying occlusion: 3D IoU evaluates shape accuracy without relying on canonical pose.

| Method | No Occlusion | | | 25% Occlusion | | | 50% Occlusion | | | 75% Occlusion | | |
|---|---|---|---|---|---|---|---|---|---|---|---|---|
| | IoU25 | IoU50 | IoU75 | IoU25 | IoU50 | IoU75 | IoU25 | IoU50 | IoU75 | IoU25 | IoU50 | IoU75 |
| NOCS Wang et al. (2019b) | 0.0 | 0.0 | 0.0 | 0.0 | 0.0 | 0.0 | 0.0 | 0.0 | 0.0 | 0.0 | 0.0 | 0.0 |
| IST-Net Liu et al. (2023) | 15.5 | 5.2 | 0.5 | 13.2 | 4.0 | 0.3 | 12.5 | 3.5 | 0.1 | 8.1 | 1.5 | 0.0 |
| HS-Pose Zheng et al. (2023) | 17.0 | 6.6 | 0.9 | 14.5 | 5.0 | 0.7 | 13.7 | 4.4 | 0.3 | 8.9 | 1.9 | 0.1 |
| GenPose++ Zhang et al. (2025b) | 21.3 | 9.4 | 1.8 | 18.1 | 7.2 | 1.3 | 17.1 | 6.3 | 0.6 | 11.1 | 2.7 | 0.1 |
| **Ours** | **42.2** | **23.1** | **3.6** | **37.3** | **17.6** | **2.0** | **31.3** | **12.2** | **1.0** | **19.1** | **4.7** | **0.2** |

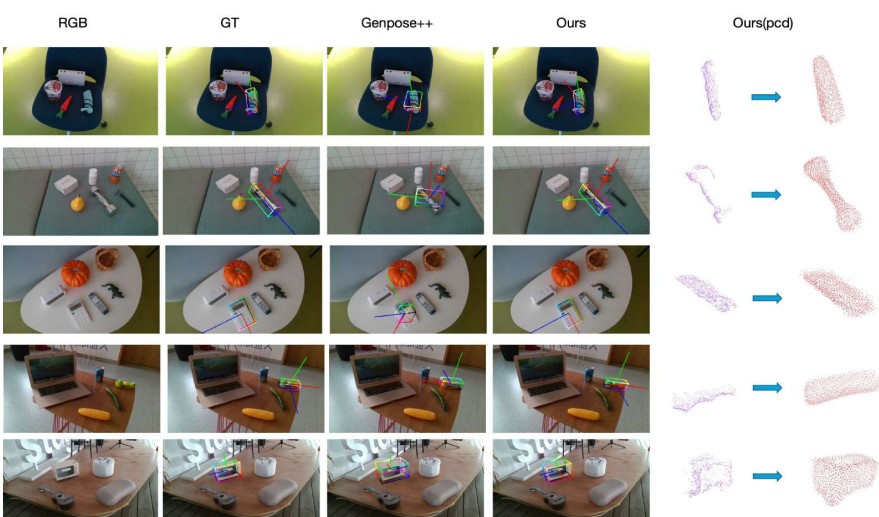

Figure 4: Qualitative results on ROPE. We show the input RGB image, ground-truth pose, poses from GenPose++ and ours, and a comparison between the predicted and ground-truth shapes.

Single-reference, where each method is given one RGB-D reference at test time. (2) Reference-free, where only Any6D and GenPose++ is applicable. Our method is reference-free in both settings.

As in Table 3, our approach outperforms both Any6D and FoundationPose under the single-reference setting, and substantially surpasses Any6D in the reference-free setting. On HANDAL, where we additionally compare against GenPose++, our method achieves consistently higher accuracy, demonstrating stronger category-level generalization. We also find that reference-based methods are sensitive to reconstruction quality: for small or irregularly shaped objects, single-view reconstructions often degrade pose predictions. In contrast, by jointly learning pose and shape directly from RGB-D input, our method delivers more robust performance across diverse unseen objects. See Fig. 5 for qualitative results.

Table 3: Quantitative comparison on the ROPE and HANDAL datasets (seen categories, unseen instances). Metrics: AUC based on 3D bounding box IoU.

| Dataset | Method | AUC ↑ | | |
|---|---|---|---|---|
| | | IoU25 | IoU50 | IoU75 |
| ROPE | FoundationPose (1 ref) | 35.0 | 18.0 | 3.1 |
| | Any6D (1 ref) | 37.2 | 19.4 | 3.5 |
| | Any6D (0 ref) | 22.5 | 8.1 | 0.3 |
| | **Ours** | **44.9** | **26.1** | **4.9** |
| HANDAL | Any6D (0 ref) | 14.5 | 3.8 | 0.0 |
| | GenPose++ (0 ref) | 16.7 | 4.3 | 0.1 |
| | **Ours** | **33.0** | **10.6** | **0.2** |

**Shape Reconstruction Performance.** Table 4 shows that our method achieves the lowest Chamfer-L1 error on both SOPE and HouseCat6D, surpassing shape-specific baselines such as AdaPointr Yu et al. (2023), Pointr Yu et al. (2021) and FoldingNet Yang et al. (2018), demonstrating strong reconstruction accuracy on both synthetic and real-world data. Unlike these baseline approaches, which rely solely on geometric cues and complex Transformer decoders, our lightweight MLP decoder yields better results. We attribute this to two factors: (i) combining RGB and depth cues for enhanced appearance–geometry representation, and (ii) unifying pose and shape estimation to introduce inherent structural constraints. These design choices lead to more complete and coherent shape reconstructions.

Table 4: Shape reconstruction on SOPE and HouseCat6D.

| Method | Chamfer-L1 ($\times 10^{-3}$) ↓ | |
|---|---|---|
| | SOPE | HouseCat6D |
| FoldingNet Yang et al. (2018) | 62.72 | 73.2 |
| Pointr Yu et al. (2021) | 29.87 | 40.2 |
| AdaPointr Yu et al. (2023) | 24.41 | 32.3 |
| **Ours** | **5.93** | **7.8** |

Table 5: Ablation study on pose estimation and shape completion (ROPE).

| Setting | AUC ↑ | | | VUS ↑ | | FPS ↑ |
|---|---|---|---|---|---|---|
| | IoU25 | IoU50 | IoU75 | 5°5cm | 10°5cm | |
| Full Model | **44.9** | **26.1** | **4.9** | **10.1** | **20.0** | 23.7 |
| Depth Only | 32.5 | 15.2 | 1.8 | 6.0 | 13.1 | **29.5** |
| w/o MoE | 41.0 | 24.0 | 3.9 | 8.7 | 17.8 | 19.2 |
| w/o Shape Completion | 38.5 | 22.2 | 3.3 | 7.9 | 16.0 | 27.8 |



Figure 5: Qualitative results on HANDAL. We compare the ground-truth 3D bounding boxes with those predicted by Any6D and our method.

Moreover, our method reconstructs the full object point cloud directly in the camera frame, making it immediately usable for downstream tasks. In contrast, other pipelines that jointly recover object pose and shape Lin et al. (2022b); Irshad et al. (2022a;b) typically predict shapes in a canonical frame and rely on separate pose estimates for camera-frame alignment. These methods produce unreliable pose estimates on our benchmark, leading to poor camera-frame reconstruction performance. Accordingly, we compare against point-cloud completion methods, which operate in the same input frame and provide fair and relevant baselines.

**Performance on reflective and transparent material.** To further assess real-world robustness on challenging materials such as reflective and transparent surfaces, which often produce unreliable or distorted depth, we separately evaluate pose estimation on ROPE and shape reconstruction on Housecat6D (Tables 6). On ROPE, we observe only a modest decrease in both pose estimation and shape reconstruction accuracy for reflective and transparent objects. We attribute this robustness primarily to the pre-trained Radio backbone, whose strong geometric priors help compensate for missing or corrupted depth—consistent with the small performance gap between depth-only and RGB-D inputs reported in Table 5. On Housecat6D, reconstruction performance on reflective objects shows only a minor drop relative to the dataset average, while accuracy on transparent objects is even slightly higher than the overall average. We attribute this to the simple and highly regular geometry of the glass objects in Housecat6D6, which makes them easier to reconstruct despite poor or incomplete depth observations.

**Ablation Study.** We conduct a series of ablation experiments on the ROPE dataset to evaluate the contribution of each major component in our framework. Results are in Table 5.

*(1) RGB–Depth Fusion.* To assess the importance of RGB guidance, we remove the RADIO encoder and use only the point cloud as input. This leads to a substantial drop in performance across all metrics, particularly in scenarios where depth observations are noisy or incomplete. This confirms dense semantic features from RGB play a crucial role in robust single-view 6D estimation. We also observe cases where RGB cues compensate for missing

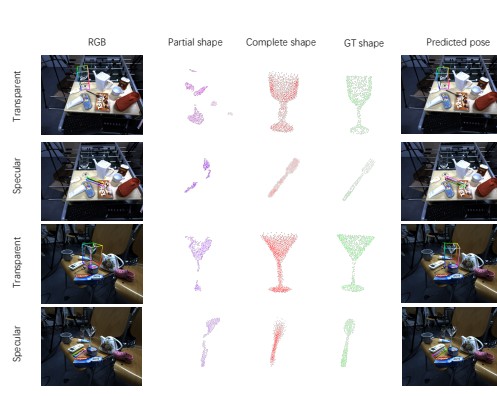

Figure 6: Specular and transparent objects from HouseCat6D.

Table 6: Performance on reflective and transparent materials across ROPE (pose estimation) and Housecat6D (shape reconstruction).

| Category | ROPE (Pose Estimation) | | | | | | Housecat6D (Shape) |
|---|---|---|---|---|---|---|---|
| | AUC@IoU25 ↑ | AUC@IoU75 ↑ | VUS@5°2cm ↑ | VUS@10°2cm ↑ | Rot Err (deg) ↓ | Trans Err (cm) ↓ | Chamfer ↓ |
| Transparent | 42.2 | 4.3 | 9.8 | 19.3 | 25.3 | 1.25 | 6.4 |
| Reflective | 41.9 | 4.2 | 9.4 | 19.2 | 25.2 | 1.35 | 10.3 |
| Average | 44.9 | 4.9 | 10.1 | 19.4 | 27.7 | 1.25 | 7.8 |

geometry in partial point clouds, enabling more accurate reconstruction of object shapes that would otherwise be ambiguous.

*(2) MoE.* We evaluate effects of MoE by replacing it with a standard Transformer feed-forward network of comparable capacity. Even with matched parameters, the model without MoE shows consistently lower accuracy, and inference becomes slower. This demonstrates that expert specialization not only improves accuracy in modeling object diversity but also accelerates inference without additional cost.

*(3) Shape Supervision.* Removing the shape reconstruction branch reduces overall accuracy and slows convergence, indicating that shape prediction serves as a strong auxiliary signal for learning robust object representations. Further analysis in Appendix 7.8 shows that our coarse-to-fine refinement and point selection mechanism also contribute positively to shape quality and pose accuracy.

These ablations validate our key design choices: RGB–depth fusion for rich visual grounding, MoE-enhanced Transformer encoding for scalable representation, and multi-task learning for improved generalization—all while maintaining real-time efficiency.

**Failure Analysis.** While our method is generally robust, several characteristic failure modes remain. (1) Reflective and transparent materials: The model typically handles such objects well, but errors occur when depth corruption affects key geometric regions (e.g., mug handles, thin metallic edges), leading to inaccurate pose estimates. (2) Heavy occlusion: Performance is stable under moderate occlusion but degrades when essential structural cues are missing, making object orientation ambiguous and reducing reconstruction completeness. (3) Category ambiguity: Symmetric, low-texture, or very small objects (e.g., books, bottles, chess pieces) often lack distinctive cues, resulting in 90°–180° pose deviations. Additional qualitative examples are provided in the appendix.

## 5 CONCLUSION

We present an end-to-end framework for joint 6D pose, size, and shape estimation from a single RGB-D image, without relying on CAD models, reference views, or category labels at inference. Our approach fuses dense semantic features from a vision foundation model with geometric point cloud data, and employs a Transformer encoder with Mixture-of-Experts (MoE) layers to improve capacity while maintaining efficiency. A multi-branch decoder enables coarse-to-fine shape reconstruction and direct pose–size regression, supporting fast and accurate 6D understanding. Our method is trained entirely on synthetic data and evaluated across four benchmarks SOPE, ROPE, ObjaversePose, and HANDAL, covering both synthetic and real-world domains, as well as seen and unseen object categories. It achieves state-of-the-art accuracy on seen instances and demonstrates strong generalization to novel objects. These results support the value of unified, reference-free inference pipelines for 6D estimation tasks.

**Future Work** While our model generalizes across diverse object types, its performance is still bounded by the coverage of training categories and may degrade on long-tail or atypical shapes underrepresented in synthetic data. Moreover, reconstructed geometries can miss fine-grained details, and the current design does not account for articulated or deformable objects. Future directions include scaling to richer corpora such as ObjaverseXL, advancing articulation and deformation modeling, and extending toward truly open-world, task-driven 6D understanding in robotics and embodied AI.

**Broader Impact.** We show that efficient regression-based models enhanced by foundation features and MoE scaling can offer strong generalization and fast inference for our 6D tasks. By eliminating the need for category priors or inference-time references, our approach may facilitate deployment in broader settings, such as robotics, augmented reality, and embodied intelligence systems.

## 6 RECOMMENDED STATEMENT

According to ICLR policy, this section is excluded from the page limit.

**Ethics Statement.** This work complies with the ICLR Code of Ethics. It does not involve human subjects or sensitive personal data. All datasets used are either publicly available or synthetically generated, and any proprietary assets are properly licensed. We are not aware of any foreseeable negative societal impact or potential misuse of the proposed method.

**Reproducibility Statement.** We are committed to ensuring the reproducibility of our results. Comprehensive details of the model architecture, training procedure, evaluation metrics, and dataset preprocessing steps are provided in the main paper, as well as in Appendix 7.1, Appendix 7.2, and Appendix 7.3. Although we do not release code at submission time, all essential implementation details are included to support independent reproduction. We also commit to releasing the code and pretrained models publicly upon publication.

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

# 7 APPENDIX

## 7.1 IMPLEMENTATION DETAILS

The proposed model jointly estimates 6D object pose, size, and shape from a single RGB-D input. For the RGB modality, we use a frozen, pre-trained RADIO-V2.5-L network to extract semantic features. Specifically, we retrieve intermediate feature maps from layers 8, 16, and 23, each with

1024 channels and a spatial resolution of $14 \times 14$. These feature maps are fused using learnable weights and processed by a lightweight convolutional block, yielding a unified 1024-channel feature map. For each 3D point, we assign a corresponding RGB feature by indexing into this fused feature map based on its 2D projection, following a strategy similar to that in Wang et al. (2019a). The input point cloud consists of 1024 points, each represented by its 3D coordinates and a 1024-dimensional RGB feature, resulting in fused inputs of shape (B, 1024, 1027). These inputs are fed into a DGCNN-based encoder with two downsampling stages ($1024 \rightarrow 512 \rightarrow 128$), where each stage includes graph convolutions and feature aggregation. The resulting 128 tokens are then passed through a geometry-aware transformer inspired by Yu et al. (2021), where the initial layer augments self-attention with a KNN-based geometric attention module. Global features are obtained via max pooling and passed through three parallel MLP heads to predict object rotation (in 6D representation), translation, and size. Additionally, the model regresses a set of candidate points from the global feature and concatenates them with a subset of the input point cloud. The ranking module selects the top 512 most confident points to form a coarse point cloud. Each coarse point is then expanded into four fine-grained points via local folding, resulting in a dense reconstruction of 2048 points.

## 7.2 Training Details

Our model is trained for a total of 50 epochs with a batch size of 128, using the AdamW optimizer. The initial learning rate is set to 1e-4, with a weight decay of 5e-4. A LambdaLR scheduler decays the learning rate by a factor of 0.9 every 8 epochs, with a minimum ratio of 2% of the initial value. We also apply a BatchNorm momentum scheduler, reducing the momentum from 0.9 by a factor of 0.5 every 3 epochs, with a lower bound of 0.01, to progressively stabilize feature normalization. All experiments are conducted on a workstation with 4× NVIDIA RTX 4080 GPUs (16 GB), an AMD EPYC 7402 24-core CPU, and 128 GB RAM. We use PyTorch 2.4 with CUDA 12.4, and enable automatic mixed-precision (AMP) and DistributedDataParallel training with synchronized BatchNorm.

## 7.3 ObjaversePose Dataset Construction

To support large-scale category-level estimation of object pose, size, and shape, we construct ObjaversePose, a synthetic RGB-D dataset derived from high-quality CAD models in ObjaverseDeitke et al. (2022). While Objaverse contains over 800,000 models, many are unsuitable for pose-related tasks due to issues such as non-watertight geometry, lack of texture, multi-object compositions, or lack a meaningful canonical orientation. We curate a clean and diverse subset through multi-stage filtering and manual processing.

### 7.3.1 CAD Model Selection and Canonical Alignment

We construct our dataset from Objaverse by intersecting two curated subsets: (1) the high-quality models filtered by LGM Tang et al. (2024), which remove low-quality geometry through caption-based and texture-based heuristics, and (2) the models with LVIS annotations, which enable fine-grained category grouping. We further discard categories with fewer than 15 high-quality instances, yielding 184 categories and 3,355 CAD models.

Each model is manually aligned to a canonical coordinate frame: the object is centered at the origin, the x-axis points forward, and the y-axis points upward, consistent with the SOPE canonical standard. In addition, we compute and annotate object-level symmetries for use in both evaluation and learning.

To assess generalization, we designate 154 tabletop-scale categories (e.g., household and office items), comprising 2,354 instances, as a held-out test set. These categories are both diverse and structurally coherent, making them well-suited for evaluating generalization.

### 7.3.2 Physically-Based Rendering with SAPIEN

We use the SAPIEN simulator to render photorealistic RGB-D data. For each model, 500 camera viewpoints are uniformly sampled from the upper hemisphere, with small perturbations added to increase diversity. Cameras are oriented toward the object center, with the z-axis pointing inward

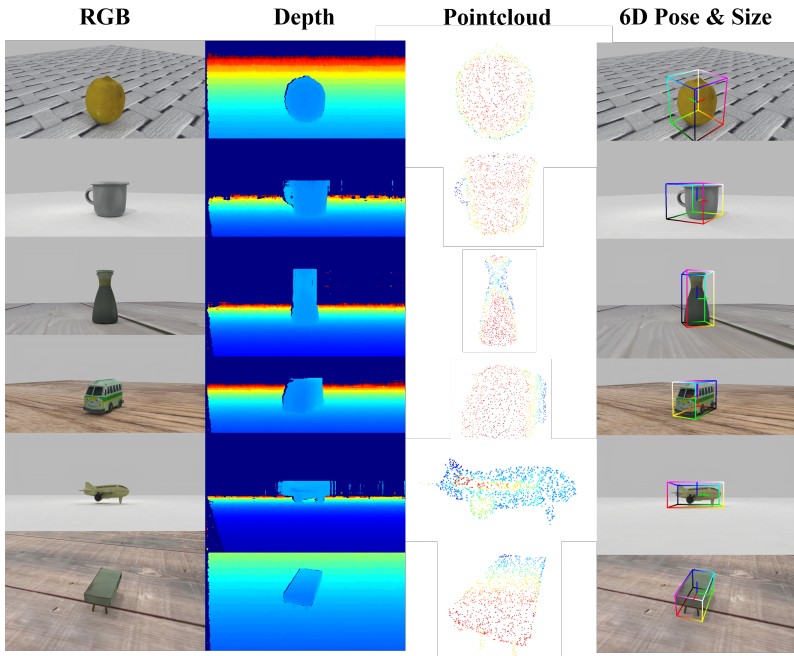

Figure 7: Examples from the Rendered ObjaversePose Dataset

and the x-axis aligned with the ground. For evaluation, we select 20 canonical views that avoid extreme top-down or side angles, ensuring consistent and balanced comparison across objects.

RGB images are rendered via ray tracing, and depth maps are generated using a physically based sensor model calibrated to the Intel RealSense D415, including matched intrinsics. From the RGB-D pairs, we compute point clouds and ground-truth object poses based on known camera–object transformations. Object textures are preserved, while ground plane textures are randomly sampled from a diverse material set. Lighting is provided by a fixed overhead point light, enriching appearance variation without introducing bias.Leveraging GPU acceleration in SAPIEN, we render 1M images in 13 hours using 8× RTX 2080 Ti GPUs. Examples are shown in Fig. 7.

We will release the full dataset—including CAD models, canonical transforms, rendered RGB-D data, and camera parameters—to support future research and benchmarking.

## 7.4 SEMANTIC AND GEOMETRIC DATASET COMPARISONS

To further clarify the differences between **Ob-javersePose/HANDAL** and **SOPE**, we perform both semantic and geometric comparisons.

**Semantic comparison.** We extract CLIP ViT-B/32 text features from all category names in both datasets and visualize their relationships using t-SNE.

**Geometric comparison.** Using canonicalized objects from each dataset, we apply a consistent normalization procedure, extract geo-

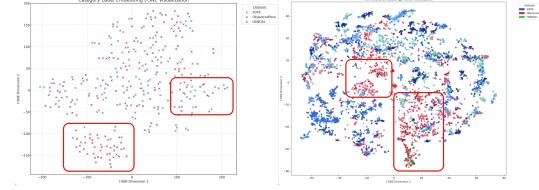

Figure 8: t-SNE visualization of semantic (CLIP) and geometric (PointNet++) features, illustrating that ObjaversePose/HANDAL covers a broader and more diverse range of categories and shapes than SOPE.

metric features using a PointNet++ model pretrained on ModelNet40 classification, and visualize them with t-SNE.

The results indicate that, while there is some overlap, these datasets exhibit distinct semantic and geometric clusters. ObjaversePose includes a broader range of categories and instances that are not

found in SOPE, suggesting notable differences between the datasets and indicating that Objaverse-Pose can serve as a useful benchmark for evaluating performance on unseen categories.

### 7.5 CONTROLLED ROBUSTNESS EVALUATION UNDER PHOTOMETRIC AND GEOMETRIC PERTURBATIONS

To quantitatively assess robustness under controlled sensing perturbations, we introduce systematic distortions across RGB and depth modalities **??**. These perturbations are designed to reflect common real-world degradations observed in deployed robotic and perception systems.

**Photometric Perturbations (RGB).** For RGB inputs, we apply controlled photometric shifts, including exposure adjustments ($\pm 0.5/\pm 1.3/\pm 2.2$ EV), white-balance changes ($\pm 800/\pm 2000/\pm 4000$ K), and synthetic illumination variations (gradient or vignetting at $0.10/0.25/0.40$ strength). As shown in Table 7, performance remains largely stable across all perturbation levels, with only minor reductions in accuracy. This insensitivity reflects the strong photometric invariances learned by the pretrained visual backbone.

**Geometric Perturbations (Depth).** To model realistic depth-sensor degradation, we inject heteroscedastic noise of the form

$$\sigma(z) = \alpha + \beta z^2,$$

where $\alpha \in \{0.001, 0.002, 0.003\}$ and $\beta \in \{0.000, 0.003, 0.006\}$. This formulation captures the distance-dependent noise of commodity depth sensors. As summarized in Table 8, performance degrades gradually with increasing noise severity, but the model retains strong accuracy even under the heavy-noise setting. While geometric perturbations have a more pronounced effect than photometric ones, the method demonstrates strong overall robustness.

Across all controlled perturbations, the method exhibits high robustness to photometric variations and stable performance under realistic depth noise, supporting its suitability for deployment in uncontrolled environments.

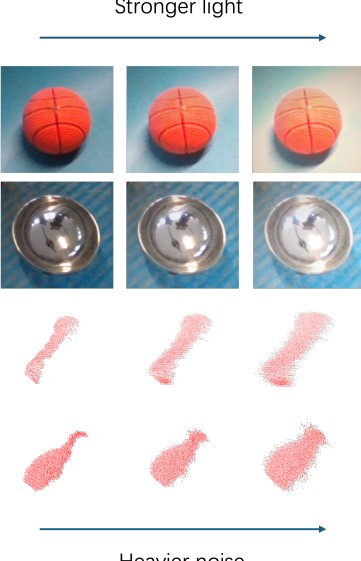

Figure 9: Controlled perturbation across rgb and depth.

### 7.6 FAILURE MODE ANALYSIS

While our main paper highlights strong qualitative performance across diverse scenarios, it is equally important to examine where the method fails. Below, we provide a detailed analysis of three primary failure modes observed on ROPE: reflective materials, severe occlusion, and category ambiguity.

Table 7: Controlled robustness under photometric perturbations on RGB.

| Perturbation | IoU25 | IoU50 | IoU75 | 5°2cm | 5°5cm | 10°2cm | 10°5cm | ΔAUC / ΔVUS (%) |
|---|---|---|---|---|---|---|---|---|
| Clean (s=0) | 44.9 | 26.1 | 4.9 | 10.1 | 14.1 | 20.0 | 29.1 | +0.0 / +0.0 |
| Light (low) | 44.7 | 26.0 | 4.8 | 10.1 | 14.1 | 19.9 | 28.8 | -0.5 / -0.6 |
| Light (moderate) | 44.5 | 25.7 | 4.7 | 10.0 | 13.9 | 19.7 | 28.5 | -1.3 / -1.6 |
| Light (heavy) | 43.1 | 24.7 | 4.5 | 9.4 | 13.4 | 18.7 | 27.1 | -4.7 / -6.4 |

Table 8: Controlled robustness under heteroscedastic depth noise.

| Perturbation | $\alpha$ | $\beta$ | IoU25 | IoU50 | IoU75 | 5°2cm | 5°5cm | 10°2cm | 10°5cm | ΔAUC / ΔVUS (%) |
|---|---|---|---|---|---|---|---|---|---|---|
| Clean (s=0) | 0.000 | 0.000 | 44.9 | 26.1 | 4.9 | 10.1 | 14.1 | 20.0 | 29.1 | +0.0 / +0.0 |
| Low | 0.001 | 0.000 | 44.7 | 26.0 | 4.8 | 10.0 | 14.1 | 20.1 | 29.2 | -0.5 / +0.1 |
| Moderate | 0.002 | 0.003 | 41.7 | 22.7 | 3.6 | 8.9 | 13.1 | 18.5 | 27.5 | -10.4 / -7.2 |
| Heavy | 0.003 | 0.006 | 37.4 | 18.3 | 2.4 | 7.3 | 11.4 | 15.8 | 24.9 | -23.4 / -20.0 |

**Reflective and transparent materials.** As shown in the main paper, our approach remains robust even when point clouds are incomplete due to transparent or reflective surfaces. We attribute this to the pre-trained Radio backbone, which supplies complementary features that mitigate missing geometric cues. However, performance degrades on highly reflective objects (e.g., kitchen knives) and slender metallic items (e.g., pens) 10, where depth corruption is extreme and minimal geometric structure is preserved.

**Occlusion.** The model performs reliably under mild to moderate occlusion, consistent with the qualitative examples presented in the supplementary video. When occlusion exceeds roughly 75%, however, too much geometric evidence is lost for the model to recover accurate poses. This results in noticeable deterioration in both rotational and translational predictions.

**Category ambiguity.** Category-ambiguity failures arise predominantly in two cases: (1) geometrically regular or strongly symmetric objects (e.g., books, flutes), where orientation cues are inherently ambiguous; and (2) small objects that occupy few pixels in RGB images and offer limited geometric detail. These factors jointly reduce the discriminative power of both visual and geometric features.

Overall, these analyses clarify the boundary conditions of our method and highlight opportunities for future work in handling extreme reflectivity, heavy occlusion, and intrinsically ambiguous object categories.

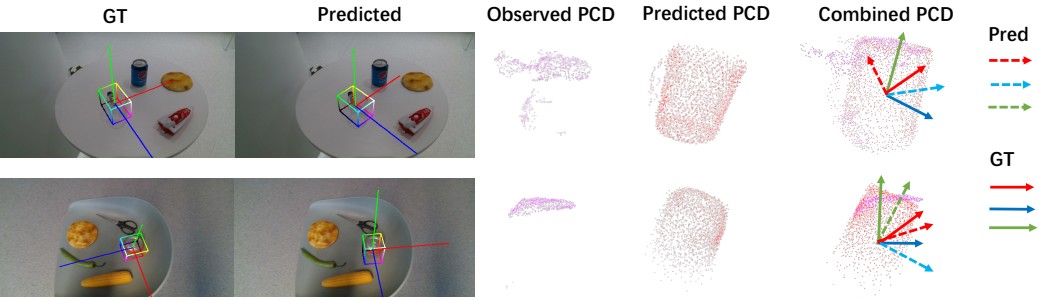

Figure 10: Failure mode analysis. Reflective surfaces on the cup handle and cup body cause incomplete or rotated point clouds, leading to corresponding pose errors.

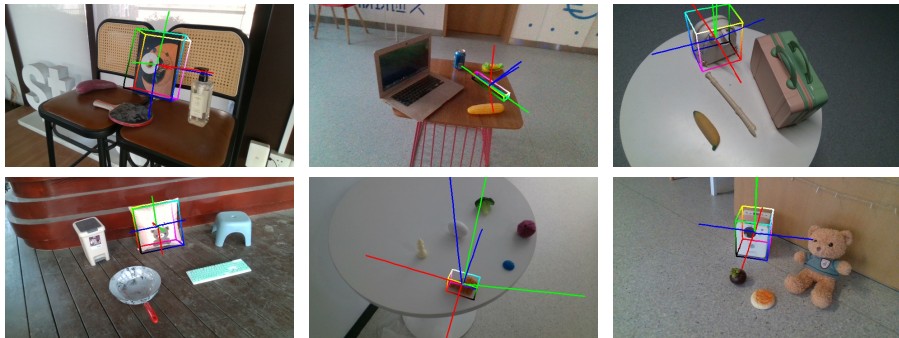

Figure 11: Failure mode analysis. Symmetric or geometrically regular objects and small objects tend to cause category ambiguity, leading to typical rotation errors of 90° or 180°.

## 7.7 EXPERT SPECIALIZATION PATTERNS

To further support the claim that the Mixture of Experts (MoE) module specializes across diverse object categories, we conducted additional analyses of expert activations. While MoE is not the primary focus of this work, the following findings provide insights into its behavior:

We analyzed expert activations across all layers, focusing on the frequency of expert selection for different object categories. The key patterns observed are:

- **Geometrically Similar Categories:** Objects with similar shapes, such as `bottle` vs. `bowl`, consistently route to the same subset of experts.

- **Coarse Structural Similarities:** Categories with partial structural overlap, like `laptop` vs. `calculator`, show partial expert overlap.

- **Geometrically Dissimilar Categories:** For highly distinct categories, such as `backpack` vs. `plug`, experts show divergent activation profiles.

We also computed cosine similarity between expert usage vectors across categories, and visualization with heatmap 12. These patterns indicate that experts naturally specialize based on object geometry, supporting the claim that MoE improves generalization across diverse object types.

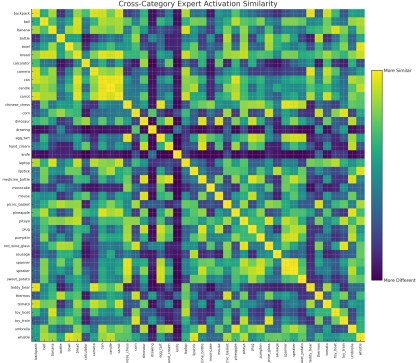

Figure 12: Expert specialization patterns.

Table 9: Consolidated ablation study results. We summarize three sets of experiments: (1) number of activated experts, (2) choice of pre-trained visual backbones, and (3) contributions of the shape reconstruction branch, coarse-to-fine strategy, and selection step.

| Setting | AUC IoU25 | AUC IoU50 | AUC IoU75 | VUS 5°2cm | VUS 10°2cm | Chamfer-L1 ($\times 10^{-3}$) |
|---|---|---|---|---|---|---|
| *Full implementation* | | | | | | |
| Ours | 44.9 | 26.1 | 4.9 | 10.1 | 20.0 | 5.93 |
| *Choice of Activated Experts* | | | | | | |
| MoE (2 in 4) | 42.8 | 24.0 | 3.2 | 8.7 | 17.3 | – |
| MoE (2 in 12) | 43.0 | 24.1 | 3.2 | 8.9 | 17.8 | – |
| MoE (1 in 8) | 40.9 | 22.5 | 2.7 | 8.1 | 16.4 | – |
| MoE (4 in 8) | 43.1 | 24.4 | 3.2 | 9.2 | 18.3 | – |
| MoE (8 in 8) | 44.2 | 25.4 | 3.7 | 9.9 | 19.5 | – |
| *Different Pre-Trained Backbones* | | | | | | |
| Replace with DINOv2 | 43.8 | 25.2 | 4.6 | 9.7 | 19.6 | – |
| Replace with CLIP | 43.1 | 24.6 | 4.3 | 9.4 | 19.1 | – |
| *Shape Reconstruction Branch, Coarse-to-Fine, and Selection* | | | | | | |
| w/o selection | 44.5 | 25.8 | 4.5 | 9.8 | 19.6 | 6.18 |
| w/o coarse-to-fine | 41.7 | 23.7 | 3.5 | 8.9 | 18.2 | 7.05 |

## 7.8 MORE ABLATION STUDY RESULTS.

21

Figure 13: Qualitative examples from ROPE

Table 9 reports a comprehensive summary of our ablation studies, covering three key aspects of our framework: (1) the number of activated experts in the MoE module, (2) the choice of pre-trained visual backbones, and (3) the contributions of the shape reconstruction branch, coarse-to-fine refinement, and selection step.

**Choice of Activated Experts.** Varying the number of activated experts shows that performance remains relatively stable across most configurations. Using only a single expert (1 in 8) leads to a clear drop in accuracy, indicating insufficient model capacity. At the other extreme, activating all experts (8 in 8) slightly improves results but incurs significantly higher computational cost. Intermediate settings (e.g., 2 in 4, 2 in 12, or 4 in 8) achieve comparable accuracy without offering clear benefits. To balance efficiency and performance, we adopt the 2-in-8 configuration, which achieves strong results with moderate computation.

**Pre-Trained Visual Backbones.** Replacing the RADIO encoder with DINOv2 or CLIP results in only minor performance degradation. This demonstrates that our framework is not tightly coupled to a specific backbone, and that the main performance gains arise from our core design rather than the choice of encoder. We therefore retain RADIO as our default backbone but note that the method remains robust with widely used alternatives.

**Shape Reconstruction, Coarse-to-Fine Strategy, and Selection.** Finally, we examine the effect of three architectural components. Removing the shape reconstruction branch degrades performance across all metrics, confirming its importance for learning shape-aware features. Eliminating the coarse-to-fine strategy produces the largest drop, with AUC IoU50 falling from 26.1 to 23.7 and Chamfer-L1 increasing from 5.93 to 7.05, suggesting that direct dense prediction fails to capture fine-grained geometry. Similarly, omitting the selection step slightly reduces accuracy and increases Chamfer-L1 due to the influence of outliers. Together, these results highlight that all three components play important and complementary roles in ensuring robust shape learning and accurate pose prediction.

## 7.9    ADDITIONAL QUALITATIVE RESULTS

We provide additional qualitative examples in Fig. 13. Each example includes the input RGB image, ground-truth annotations, predictions from the state-of-the-art baseline (GenPose++), and our model's predictions for comparison.

## 7.10    LLM USAGE.

We used ChatGPT (GPT-5) for grammar and wording refinement; all research ideas and results are by the authors.

