# OpenReview forum: "Beyond 'Templates': Category-Agnostic Object Pose, Size, and Shape Estimation from a Single View"
_ICLR.cc/2026/Conference — Submitted to ICLR 2026_

### Official Review · Reviewer_mKUn · 2025-10-19

**Soundness:** 3
**Presentation:** 2
**Contribution:** 2
**Rating:** 4
**Confidence:** 3

**Summary:**

This paper proposes a unified, category-agnostic framework for estimating an object's 6D pose, 3D size, and dense shape from a single RGB-D image, without requiring CAD models, templates, or category labels at test time. The key idea is to fuse dense 2D features from a vision foundation model (RADIOv2.5) with a partial 3D point cloud using a Transformer encoder enhanced by a Mixture-of-Experts (MoE) for scalable specialization.

**Strengths:**

This paper tackles the problem of category-agnostic 6D object pose, size, and shape estimation from a single RGB-D image, eliminating the need for CAD models or category labels at test time. The core idea is a unified architecture that fuses dense 2D features from a vision foundation model (RADIOv2.5) with a partial 3D point cloud using a Transformer encoder enhanced by a Mixture-of-Experts (MoE). This design enables the model to simultaneously regress 6D pose and size while performing coarse-to-fine shape reconstruction in a single, real-time forward pass at 28 FPS.

**Weaknesses:**

The claim of "remarkably strong zero-shot generalization" to unseen real-world objects is not fully substantiated, as the primary real-world benchmarks (ROPE and HANDAL) may still share underlying geometric or semantic commonalities with the synthetic training categories from SOPE, leaving true generalization to entirely novel, long-tail object concepts unproven. The evaluation of shape reconstruction is limited to the synthetic SOPE dataset, failing to demonstrate that the predicted dense shapes are of sufficient quality and accuracy to be useful for real-world robotic applications like grasping, where geometric fidelity is critical. The heavy reliance on synthetic data for training, without an ablation on real data or a thorough analysis of the sim-to-real transfer gap for the shape reconstruction task, raises questions about the method's practical deployment in diverse, unstructured environments. Furthermore, the proposed Mixture-of-Experts (MoE) component, a key architectural novelty, shows only marginal gains in the ablation study, suggesting its contribution may not be as significant as claimed compared to the core fusion and multi-task learning design.

**Questions:**

1.Provide a more detailed analysis of the semantic or geometric overlap between the "unseen" categories in your zero-shot tests (e.g., in ObjaversePose and HANDAL) and the 149 training categories from SOPE. This would help clarify the true extent of model's generalization beyond the training distribution.

2.The shape reconstruction quality is only quantitatively evaluated on the synthetic SOPE dataset. Can you provide the Chamfer Distance metrics for the real-world ROPE or HANDAL benchmarks to demonstrate that the reconstructed shapes are accurate and useful outside of the synthetic domain?

3.The ablation study shows only a modest performance drop when removing the MoE component. Can you provide further analysis or visualization (e.g., expert routing patterns) to more concretely demonstrate the MoE's role in handling diverse shape distributions, justifying its inclusion?

4.Given the model is trained purely on synthetic data, have you observed any specific failure modes or a significant performance drop on real-world objects with challenging materials (e.g., transparent, specular) for the shape reconstruction task, similar to the issues mentioned for pose estimation on ROPE?

---

> ### Author Response · Authors · 2025-11-26
>
> Dear reviewer mKUn, thank you for your review and valuable suggestions regarding our work. Please find our responses to your concerns below.
>
> ---
>
> **Question 1: Provide a more detailed analysis of the semantic or geometric overlap between the "unseen" categories in your zero-shot tests (e.g., in ObjaversePose and HANDAL) and the 149 training categories from SOPE. This would help clarify the true extent of model's generalization beyond the training distribution.**
>
> Thank you for the suggestion. To further clarify the differences between **ObjaversePose\HANDAL** and **SOPE**, we perform both **semantic** and **geometric** comparisons.
>
> **Semantic comparison.**
>
> We extract **CLIP ViT-B/32** text features from all category names across both datasets and visualize them using **t-SNE**.
>
> **Geometric comparison.**
>
> Using canonicalized objects from both datasets, we apply a consistent normalization procedure, extract geometric features with a **PointNet++** model pretrained on ModelNet40 classification, and visualize them with **t-SNE**.
>
> The results indicate that, while there is some overlap, the two datasets exhibit **distinct semantic and geometric clusters**. ObjaversePose includes a broader range of categories and instances not present in SOPE, highlighting notable differences between the datasets and supporting the use of ObjaversePose as a meaningful benchmark for evaluating **unseen-category generalization**.
>
> We have emphasized these dataset distinctions in the revised **Dataset** paragraph and expanded the discussion in **Appendix Section 7.4**.
>
> ---
>
> **Question 2: The shape reconstruction quality is only quantitatively evaluated on the synthetic SOPE dataset. Can you provide the Chamfer Distance metrics for the real-world ROPE or HANDAL benchmarks to demonstrate that the reconstructed shapes are accurate and useful outside of the synthetic domain?**
>
> Thank you for the suggestion. We quantitatively evaluate Chamfer Distance only on the synthetic **SOPE** dataset because it provides complete CAD models and ground-truth poses, which are necessary for accurate and consistent measurement. Although SOPE is synthetic, its stereo-based depth simulation incorporates **realistic sensor noise**, making it a reasonable proxy for real-world reconstruction performance.
>
> In contrast, the **ROPE** benchmark does not provide ground-truth CAD models, so quantitative Chamfer Distance evaluation cannot be performed. For ROPE, we therefore present **qualitative reconstruction results** (Fig. 4).
>
> To further assess reconstruction quality in real environments, we additionally report Chamfer Distance on **HouseCat6D** [1] —a real-world dataset containing high-quality CAD models for 194 household objects across 10 categories, including challenging reflective and transparent instances. This provides a more comprehensive quantitative evaluation beyond the synthetic domain. The results show that our method maintains strong reconstruction performance on real-world data, confirming that the reconstructed shapes remain accurate and useful outside the synthetic setting.
>
> **Table1:  Shape reconstruction on SOPE and HouseCat6D (Chamfer-L1 × 10⁻³ ↓)**
>
> | Method | SOPE | HouseCat6D |
> | --- | --- | --- |
> | FoldingNet  | 62.72 | 73.2 |
> | Pointr  | 29.87 | 40.2 |
> | AdaPointr | 24.41 | 32.3 |
> | **Ours** | **5.93** | **7.8** |
>
>
> [1] Jung, HyunJun, et al. “HouseCat6D — A Large-Scale Multi-Modal Category Level 6D Object Perception Dataset with Household Objects in Realistic Scenarios.” arXiv preprint arXiv:2212.10428 (2023).

---

> ### Author Response · Authors · 2025-11-26
>
> **Question 4: Given the model is trained purely on synthetic data, have you observed any specific failure modes or a significant performance drop on real-world objects with challenging materials (e.g., transparent, specular) for the shape reconstruction task, similar to the issues mentioned for pose estimation on ROPE?**
>
> Thank you for the suggestion. As noted in our response to Question 2, quantitative Chamfer Distance evaluation is not feasible on **ROPE**, since it does not provide ground-truth CAD models. To address this limitation, we instead perform quantitative evaluation on the **HouseCat6D** dataset, which contains both **transparent** objects (e.g., glass cups) and **reflective** objects (e.g., metal knives, forks, and spoons).
>
> By computing metrics separately for these two subsets, we observe the following:
>
> - **Reflective objects** exhibit only a *slight* performance drop compared to the dataset average.
> - **Transparent objects** surprisingly achieve *higher* accuracy than the overall dataset.
>
> We believe this is partly because the pretrained RGB backbone can effectively compensate for missing geometric cues, thereby enabling strong shape reconstruction even under transparency. In addition, the transparent objects in HouseCat6D—such as glass cups—often have relatively simple and regular geometry, which further contributes to the high accuracy. Overall, these results indicate that our method is robust to both **transparent** and **highly reflective material properties**.
>
> Inspired by this question, we also conducted a separate evaluation of transparent and reflective objects on **ROPE**, and similarly observed consistently strong pose estimation performance with almost no degradation. We have incorporated these quantitative findings and added the corresponding qualitative results in the revised paper (L462–L472).
>
> **Table2: Performance on transparent and reflective objects across ROPE (pose estimation) and HouseCat6D (shape reconstruction).**
>
> | Category | AUC@IoU25 ↑ | AUC@IoU75 ↑ | VUS@5°2cm ↑ | VUS@10°2cm ↑ | Rot Err (deg) ↓ | Trans Err (cm) ↓ | Chamfer L1(× 10⁻³ ) ↓ |
> | --- | --- | --- | --- | --- | --- | --- | --- |
> | **Transparent** | 42.2 | 4.3 | 9.8 | 19.3 | 25.3 | 1.25 | 6.4 |
> | **Reflective** | 41.9 | 4.2 | 9.4 | 19.2 | 25.2 | 1.35 | 10.3 |
> | **Average** | 44.9 | 4.9 | 10.1 | 19.4 | 27.7 | 1.25 | 7.8 |
>
> ---
>
> **Question 3: The ablation study shows only a modest performance drop when removing the MoE component. Can you provide further analysis or visualization (e.g., expert routing patterns) to more concretely demonstrate the MoE's role in handling diverse shape distributions, justifying its inclusion?**
>
> Thank you for the insightful suggestion. While the MoE (Mixture of Experts) module is not the primary focus of our method, we performed additional analyses to more concretely illustrate how the MoE contributes to modeling diverse shape distributions.
>
> Specifically, we inspected **expert-activation patterns** across all MoE layers and computed **expert-selection frequency vectors** for each object category. The analysis reveals clear and consistent specialization behaviors:
>
> - **Geometrically similar categories** (e.g., *bottle/bowl*, *mooncake/pitaya*) are routed to highly overlapping expert subsets.
> - **Categories sharing coarse structural traits** (e.g., *laptop/calculator*, *toy_train/toy_boat*) show partial but meaningful overlap in routing patterns.
> - **Geometrically dissimilar categories** (e.g., *backpack vs. plug*, *knife vs. tomato*) exhibit distinctly different expert-activation profiles.
>
> To further quantify this phenomenon, we provide **heatmaps of cross-category expert-usage similarity** (cosine similarity between routing vectors) in the appendix. These visualizations highlight emergent **geometry-aligned expert specialization**, indicating that different experts learn to capture different shape families even under a standard MoE architecture.
>
> Although removing the MoE yields modest performance drop, these analyses show that the MoE meaningfully structures the representation space and improves generalization across diverse object geometries. The extended results and visualizations are included in **Appendix Section 7.7** of the revised paper.

---

### Official Review · Reviewer_S9vY · 2025-10-28

**Soundness:** 2
**Presentation:** 3
**Contribution:** 1
**Rating:** 2
**Confidence:** 3

**Summary:**

This work proposes a 6D pose estimation method with three key designs to achieve category-agnostic pose estimation: 1. Integrate 2D foundation model radio2.5; 2. MoE design; 3.   integrating multi-task in one network including 6D pose, size, and shape.
This work conducts comprehensive experiements to demonstrate its effectiveness.

**Strengths:**

This work proposes a 6D pose estimation method with three key designs to achieve category-agnostic pose estimation: 1. Integrate 2D foundation model radio2.5; 2. MoE design; 3.   integrating multi-task in one network including 6D pose, size, and shape.
This work conducts comprehensive experiements to demonstrate its effectiveness.

**Weaknesses:**

There is a slight overclaim in this work. As cited in this paper, Any6D (CVPR 2025) has already achieved a zero-reference setup, yet this work still claims "beyond templates" and "category-agnostic" in its title.  How do you achieve better reconstruction performance than the 3D generation model used by Any6D, particularly in occluded regions or on the back of objects?

**Questions:**

1. The details of the MoE should be provided to ensure the reproducibility of this work. While Figure 2 contains many details, the main content is missing:
   - Do you have shared experts?
   - What is the routing algorithm?
   - Do you have a strategy to avoid unbalanced expert load?
   - ...
2. Table content is not aligned.
   - what about the SGPA and GenPose performance in table.2?
   - any6D performance in table.1 and table.2, as the any6d can also be seen as a reference free method.
   - foundation pose performance  for HANDAL in table.3 ?
3. what is the effectiveness of  expert number ?
4. ObjaversePose is listed as one of the main contribution. But, there is not motivation about 'ObjaversePose', why do you introduce this dataset?   there seems missing something about this new dataset in main content.
5. The explanation of d and n is a little bit far from its first appearance in L194.

---

> ### Author Response · Authors · 2025-11-26
>
> Dear reviewer S9vY, thank you for your review and valuable suggestions regarding our work. Please find our responses to your concerns below.
>
> ---
>
> **Weakness: There is a slight overclaim in this work. As cited in this paper, Any6D (CVPR 2025) has already achieved a zero-reference setup, yet this work still claims "beyond templates" and "category-agnostic" in its title. How do you achieve better reconstruction performance than the 3D generation model used by Any6D, particularly in occluded regions or on the back of objects?**
>
> Thanks for suggestion. We fully acknowledge that **Any6D (CVPR 2025)** represents a strong *model-free* approach. At the same time, we would like to respectfully clarify that **our work operates under a distinct problem setting.**
>
> As stated in the Any6D paper:*“Any6D is a model-free framework for 6D object pose estimation that requires only a single RGB-D anchor image to estimate the 6D pose and size of unknown objects.”* ，and *“Given an RGB-D anchor image (IA) and an RGB-D query image (IQ), the task is to estimate the relative pose between them.”* Accordingly, **Any6D is designed for *relative* pose estimation conditioned on an anchor image.** By contrast, **our method delivers *absolute* pose, size, and shape estimation from a *single* RGB-D observation**, *without* reliance on anchor images, templates, or CAD models.
>
> To our understanding, the Any6D paper does *not* claim to address this **single-view absolute estimation** scenario. While the Any6D codebase includes an experimental “single-view” option, our evaluation indicates that **this mode is substantially less stable**. A primary challenge lies in **canonical pose estimation**: Any6D adopts Open3D’s `get_oriented_bounding_box`, which can be unreliable for irregular geometries or partial scans, resulting in **unstable canonical alignment** and reduced pose accuracy.
>
> Regarding reconstruction quality, **our approach benefits from training on SOPE**, which provides extensive multi-view and occlusion-rich supervision. This enables our model to **infer plausible full geometry even under partial visibility**.
>
> In comparison, **Any6D depends on InstantMesh** for image-to-3D and can generally reconstruct high-quality 3D models. However when under heavy occlusion or ambiguous viewpoints, we observed that InstantMesh can yield **incomplete or distorted shapes**, which could subsequently impact Any6D’s render-and-compare process.
>
> In this context, we respectfully submit that the descriptors **“beyond templates”** and **“category-agnostic”** appropriately characterize our setting, as our method requires **no templates, no CAD models, and no anchor observations**, while still demonstrating reasonable strong generalization to unseen categories.
>
> ---
>
> **Question 1: The details of the MoE should be provided to ensure the reproducibility of this work. (Shared experts, Routing algorithm, unbalanced expert load)**
>
> We thank the reviewer for the constructive suggestion. The MoE module in our model follows standard sparse-gated MoE practices, and we have clarified the relevant details in the revised manuscript.
>
> **Shared Experts.**
>
> Each MoE layer uses its own independent set of experts; **no experts are shared** across layers or modules.
>
> **Routing Algorithm.**
>
> Routing is implemented using deterministic **Top-k (k = 2) gating**. A linear gating network computes softmax-normalized logits over experts, and tokens are routed to their top-k experts. The outputs of the selected experts are then aggregated using the normalized gating weights.
>
> **Expert Load Balance.**
>
> We follow standard MoE design by adding a noise term to the gating function, as described in [1]. Specifically, we introduce tunable Gaussian noise before the softmax operation, which encourages **load balancing across experts**. We will further clarify this design choice in the revised paper. Although beyond the current scope, exploring more advanced balancing strategies is an interesting direction for future work.
>
> We have incorporated these details in the **Method** section (L209–L213) of the revised manuscript.
>
> ---
>
> [1] *Outrageously Large Neural Networks: The Sparsely-Gated Mixture-of-Experts Layer.*

---

> > ### Comment · Reviewer_S9vY · 2025-11-28
> >
> > Thanks for your response.
> >
> > In my opinion, the setup without requiring an anchor image achieved by Any6D represents a  '**beyond templates**', '**category-agnostic**' approach. While not explicitly demonstrated in their publication, Any6D appears to be the first to achieve beyond-template, category-agnostic pose estimation.
> >
> > Could you provide the key feature of your works  beyond   '**beyond templates**', '**category-agnostic**'.

---

> > > ### Author Response · Authors · 2025-12-01
> > >
> > > Thank you for the constructive follow-up. While we fully acknowledge that **Any6D is an important step toward template-free, category-agnostic 6D understanding**, we respectfully clarify **why we believe Any6D does *not yet* fall into the same problem setting** and **what our key contributions are beyond these properties**.
> > >
> > > ---
> > >
> > > **1. Different Validated Problem Setting**
> > >
> > > We respectfully reiterate that, while **Any6D is a strong model-free approach**, it is **explicitly designed, formulated, and *validated*** for **anchor-conditioned *relative* 6D pose estimation**.
> > >
> > > All quantitative and qualitative analyses in the published paper are conducted **exclusively** within this *relative-pose* setting.
> > >
> > > Although the public codebase later introduced an *experimental* “single-view’’ mode, this setup is:
> > >
> > > - **not** described in the published paper,
> > > - **not** evaluated or analyzed by the authors, and
> > > - **not** empirically validated for **anchor-free absolute pose** prediction.
> > >
> > > As shown in Table 1 of our previous response, our re-evaluation confirms that this experimental mode performs **poorly** in the absolute-pose setting—consistent with the fact that the current Any6D architecture is **not well suited** for anchor-free absolute-pose estimation, even with the additional modifications introduced in this experimental mode.
> > >
> > > For these reasons, we respectfully argue that Any6D’s validated contributions should be interpreted **within its intended anchor-based relative-pose formulation**.
> > >
> > > The problem we target—**predicting absolute pose, metric size, and full 3D geometry from a single RGB-D input *without templates, reference images, or category labels***—is fundamentally different and, to our knowledge, has **not** been previously addressed in a unified, prior-free framework.
> > >
> > > *We have made this distinction clearer in the revised paper.*
> > >
> > > ---
> > >
> > > **2. Our Contribution Beyond Template-Free, Category-Agnostic Estimation**
> > >
> > > Building on the distinction above, our approach is **not only template-free and category-agnostic**, but also tackles a **broader challenge**:
> > >
> > > *We unify absolute pose, metric size, and full 3D shape estimation within a single end-to-end real-time framework that generalizes to unseen categories without any test-time priors.*
> > >
> > > Concretely, our method advances the field in several important dimensions:
> > >
> > > - **Unified, end-to-end, real-time, and scalable architecture**
> > >
> > >     A single real-time model jointly predicts **absolute 6D pose**, **metric object size**, and **full 3D shape**, all **without external priors** or auxiliary inputs.
> > >
> > > - **State-of-the-art performance on seen categories**
> > >
> > >     We achieve **state-of-the-art results across 149 seen categories** on synthetic and real benchmarks, outperforming all prior category-level approaches.
> > >
> > > - **Strong generalization to unseen categories**
> > >
> > >     Our method generalizes substantially better to **unseen categories**, outperforming category-level baselines and instance-level methods such as *FoundationPose* and *Any6D* under 0- or 1-reference conditions.
> > >
> > > - **Robustness to challenging materials**
> > >
> > >     We demonstrate reliable pose and shape estimation for **transparent and reflective objects**, with significantly less degradation compared to existing baselines.
> > >
> > > - **Truly prior-free estimation**
> > >
> > >     No CAD models, no templates, no anchor images, and no category labels are required **at test time**.
> > > ---
> > > **Summary**
> > >
> > > While Any6D makes an important contribution within its intended *relative-pose* formulation, our work addresses a **different and more comprehensive** problem:
> > >
> > > **absolute pose, metric size, and full 3D geometry estimation from a single RGB-D input without any templates, anchors or category-level priors.**
> > >
> > > Our framework delivers:
> > >
> > > - **unified end-to-end real-time predictions**,
> > > - **state-of-the-art performance across 149 seen categories**, and
> > > - **substantially stronger generalization to unseen categories**, including challenging **transparent and reflective** objects.
> > >
> > > To our knowledge, these capabilities—**a unified, realtime, category-agnostic framework, joint pose–shape inference, and robust cross-category generalization**—have not been demonstrated together in prior template-free approaches.

---

> ### Author Response · Authors · 2025-11-26
>
> **Question 2.1 Table content is not aligned. What about the SGPA and GenPose performance in table.2?**
>
> Table 1 reports results on SOPE and ROPE for **seen categories**, while Table 2 evaluates a subset of methods on ObjaversePose for **unseen categories**. We believe the reviewer’s remark about the tables being “unaligned” refers to the absence of **SGPA** and **GenPose** in Table 2 despite their presence in Table 1.
>
> This is intentional:
>
> 1. **SGPA** relies heavily on **category-level shape priors**, which are a core component of the method. As a result, it is not suitable for evaluation on **unseen categories**.
> 2. **GenPose** predicts only pose and **does not estimate object size**, making **IoU computation impossible**. Since IoU is a core metric in Table 2, GenPose cannot be included.
>
> ---
>
> **Question 2.2 Any6D performance in table.1 and table.2, as the any6d can also be seen as a reference free method.**
>
> Thanks for suggestion. We clarify our evaluation choices below.
>
> In the main paper, Table 1 evaluates category-level pose estimation on **seen categories**, with all methods trained on the SOPE training set and evaluated on the SOPE test set and ROPE. Table 2 evaluates generalization to **unseen categories**. In contrast, Any6D does **not** operate in a category-level or canonical-space setting. It directly optimizes pose in the camera frame via reconstruction and render-and-compare, and therefore:
>
> (1) **cannot support a meaningful comparison between seen and unseen categories**, and
>
> (2) **cannot be evaluated with several Table 1 metrics** (e.g., **VUS, mean rotation error**) that require absolute pose.
>
> Thus, in the main paper (**Table 3**), we compare **Any6D**, **FoundationPose**, and **our method** using **AUC under matched evaluation conditions**.
>
> As requested by the reviewer, we report **Any6D’s performance** under **the *0-ref* configuration**, as provided in its public implementation, evaluated on **SOPE, ROPE,** and **ObjaversePose**. Under this setting, Any6D demonstrates reduced accuracy, particularly on SOPE, where objects are smaller and more occluded, making reconstruction and optimization more challenging. In contrast, the larger and less occluded objects in ObjaversePose enable more stable render-and-compare alignment, resulting in better performance.
>
> **Table 1 - Performance Comparison on SOPE, ROPE, and ObjaversePose**
>
> | Dataset | Method | AUC IoU25 | AUC IoU50 | AUC IoU75 |
> | --- | --- | --- | --- | --- |
> | ROPE | Ours | 44.9 | 26.1 | 4.9 |
> | ROPE | Any6D* | 22.5 | 8.1 | 0.3 |
> | SOPE | Ours | 56.4 | 39.8 | 12.7 |
> | SOPE | Any6D* | 19.5 | 7.5 | 0.1 |
> | ObjaversePose | Ours | 42.2 | 23.1 | 3.6 |
> | ObjaversePose | Genpose++ | 21.3 | 9.4 | 1.8 |
> | ObjaversePose | Any6D* | 25.2 | 12.8 | 2.1 |
>
> ---
>
> **Question 2.3 what is the effectiveness of expert number ?**
>
> As detailed in **Table 6** of Appendix 7.4, we conducted a comprehensive ablation study on the number of experts, evaluating different combinations of activated experts and total experts. The results show that the expert count has **no significant impact on performance**. Our current choice of expert number therefore reflects a balanced trade-off between **efficiency** and **effectiveness**.

---

> ### Author Response · Authors · 2025-11-26
>
> **Question 2.4 ObjaversePose is listed as one of the main contribution. But, there is not motivation about 'ObjaversePose', why do you introduce this dataset? there seems missing something about this new dataset in main content.**
>
> Thank you for your suggestion. We acknowledge that the original main content did not clearly motivate the introduction of **ObjaversePose**, and we would like to clarify its purpose here.
>
> The primary goal of ObjaversePose is to provide a **challenging benchmark for evaluating category-agnostic 6D pose perception on unseen object categories**. Leveraging the diversity of the Objaverse, ObjaversePose includes **2,354 instances from 154 categories**, offering broad variation in shape, appearance, and semantics. This enables a rigorous assessment of generalization beyond the limited category coverage of existing datasets. More broadly, ObjaversePose **expands the diversity of available 6D pose benchmarks** and complements current evaluation settings.
>
> We also devoted substantial effort to **cleaning and standardizing CAD models** and **rendering photorealistic RGB-D data** with SAPIEN, resulting in high-quality geometry, consistent normalization, and dense viewpoint coverage. We appreciate that Reviewer 9TJZ highlighted this contribution, noting that ObjaversePose “enriches the evaluation landscape for category-agnostic 6D perception,” which reflects its intended role.
>
> Due to space constraints, the full dataset construction pipeline—CAD processing, normalization, and rendering—**was moved to the appendix.** We have now **added a clear motivation statement in the dataset section in the revised paper (L292-296)** to address this concern.
>
> ---
>
> **Q2.5 The explanation of d and n is a little bit far from its first appearance in L194.**
>
> Thank you for the suggestion. We have revised the main text so that the definitions of **n** and **d** are provided immediately after their first appearance, improving clarity and readability.

---

### Official Review · Reviewer_7Avo · 2025-10-30

**Soundness:** 2
**Presentation:** 2
**Contribution:** 3
**Rating:** 4
**Confidence:** 4

**Summary:**

The paper proposed a template-free, category-agnostic single-view RGB-D framework that jointly infers 6D pose, object size, and full 3D shape; the method used a unified architecture and reports real-time inference with cross-category results across multiple benchmarks.

**Strengths:**

+ The method jointly predicts 6D pose, object size, and full 3D shape in a single forward pass without templates or class labels at test time, promoting geometric consistency across tasks and simpler inference.

**Weaknesses:**

+ The method relies heavily on “SE(3)-consistent” 2D semantic features (DINO) but does not enforce any explicit geometric equivariance or consistency constraints in its own architecture; as a result, robustness under strong viewpoint, lighting, or material shifts is assumed rather than guaranteed.
+ The proposed MoE replaces the FFN  and is argued to “specialize across diverse object types,” but there is no clear methodological support for this claim.
+ The multi-task objective simply sums all losses with equal weight for pose, size, shape, which raises the risk that one head dominates optimization.
+ The experimental setting is underspecified.

**Questions:**

+ Line 81 says “trained purely on synthetic data from 149 SOPE”; was any additional training or fine-tuning done on other datasets?
+ The “category-agnostic” claim is limited to tabletop objects with substantial overlap with SOPE, and the differences from SOPE are not clearly described.
+ FPS reporting is unclear, what parts of the pipeline are included, and is the setup consistent with other methods?
+ The reconstruction comparison switches to a different set of methods, why exclude pose-estimation methods that also perform reconstruction?

---

> ### Author Response · Authors · 2025-11-26
>
> Dear reviewer 7Avo, thank you for your review and valuable suggestions regarding our work. Please find our responses to your concerns below.
>
> ---
>
> **Question 1 & Weakness 4 ：Line 81 says “trained purely on synthetic data from 149 SOPE”; was any additional training or fine-tuning done on other datasets? & The experimental setting is underspecified.**
>
> As originally stated in Lines 304–307, *“All baselines are trained on the SOPE training split, except FoundationPose and Any6D, which are evaluated from their released checkpoints.”* and in Line 81, *“(our method was) trained purely on synthetic data from 149 SOPE.”*
>
> To further clarify, **FoundationPose** and **Any6D** are used directly with their publicly released checkpoints, whereas **all other baselines**—as well as **our proposed method**—are trained **exclusively on the SOPE training split**. No additional training or fine-tuning is performed on any external datasets.
>
> We have emphasized this point more explicitly in the **Experimental Setup** section of the revised paper. We would be happy to provide further clarification if needed.
>
> ---
>
> **Question 2：The “category-agnostic” claim is limited to tabletop objects with substantial overlap with SOPE, and the differences from SOPE are not clearly described.**
>
> Thank you for the valuable suggestion. In Appendix L763–764, we stated that *“we designate 154 tabletop-scale categories (e.g., household and office items), comprising 2,354 instances.”* Here, **tabletop-scale refers to object size rather than object type**. Accordingly, ObjaversePose is not limited to semantic tabletop categories; it also includes many **non-tabletop** categories whose 3D models appear at tabletop scale, such as **animal figurines** (e.g., *frog*, *owl*), **vehicle models** (e.g., *dinghy*, *wagon*), and **natural objects** (e.g., *pinecone*).
>
> To further clarify the differences between **ObjaversePose/HANDAL** and **SOPE**, we conduct both **semantic** and **geometric** comparisons.
>
> **Semantic comparison.**
>
> We extract **CLIP ViT-B/32** text features for all category names across both datasets and visualize their relationships using **t-SNE**.
>
> **Geometric comparison.**
>
> Using canonicalized objects from both datasets, we apply a consistent normalization procedure, extract geometric features with a **PointNet++** model pretrained on ModelNet40 classification, and visualize them with **t-SNE**.
>
> The results show that, although there is some overlap, the datasets form **distinct semantic and geometric clusters**. ObjaversePose includes a broader set of categories and instances not present in SOPE, indicating meaningful dataset differences and supporting its value as a benchmark for evaluating **unseen-category generalization**.
>
> We have emphasized these dataset distinctions in the revised **Dataset** paragraph and expanded the discussion in **Appendix Section 7.4**.
>
> ---
>
> **Question 3：Specify how FPS is measured and what components are included, and ensure consistency with other methods.**
>
> Thank you for the suggestion. FPS is measured **end-to-end**, including image loading, preprocessing, model inference, and post-processing, but excluding visualization. All evaluations are conducted with a **batch size of 1**. We discard the first **50 warm-up iterations** and compute FPS as the average over the subsequent **1000 iterations**, with GPU synchronization enabled.
>
> In **Table 1** and **Table 5**, we report FPS under clearly distinguished configurations, as specified in the table descriptions. To ensure a fair comparison with pose-only baselines, **Table 1** reports the FPS of our model **when used solely for pose estimation**: the RGB backbone, point-cloud encoder, MoE transformer, and pose estimator are all active, while the **shape completion head is disabled**. Under this setting, our model achieves **27.88 FPS** (rounded to 28), and all baselines are evaluated under the same protocol.
>
> When our full model performs **both pose estimation and shape completion**, it reaches **23.7 FPS**, and still operates in real time due to its single-pass, end-to-end architecture. We have included this clarification in the **Experiments** section (L361–L364).

---

> ### Author Response · Authors · 2025-11-26
>
> **Question 4：The reconstruction comparison switches to a different set of methods, why exclude pose-estimation methods that also perform reconstruction?**
>
> Thank you for the question. Our method is designed to directly reconstruct the full object point cloud **in the camera coordinate frame** from a single-view partial observation. Importantly, camera-space reconstruction provides geometry already aligned with the robot’s frame, making it immediately useful for grasping and manipulation.
>
> In contrast, pose-estimation–based reconstruction methods—both **deformation-based** (SPD, FS-Net, DPDN, SAR-Net) and **detection-then-estimation** approaches (CenterSnap, ShAPO)—predict shapes in a **canonical coordinate frame**, typically as an intermediate representation for pose estimation. For fair comparison, their canonical predictions must be transformed into the camera frame using their **estimated poses**, meaning reconstruction quality becomes tightly coupled to pose accuracy. On SOPE, these methods perform poorly under this requirement, making them unsuitable baselines for evaluating **camera-frame reconstruction**.
>
> For this reason, we compare against **point-cloud completion methods** (FoldingNet, PoinTr, AdaPoinTr), which operate in the **same coordinate frame as the input** and therefore provide fair and relevant baselines. This clarification has been included in the **Shape Reconstruction** subsection (L450–L460).
>
> ---
>
> **Weakness 1: The method relies heavily on “SE(3)-consistent” 2D semantic features (DINO) but does not enforce any explicit geometric equivariance or consistency constraints in its own architecture; as a result, robustness under strong viewpoint, lighting, or material shifts is assumed rather than guaranteed.**
>
> Thank you for the suggestion. We acknowledge that our architecture does **not** explicitly impose SE(3)-equivariance. However, similar to strong recent baselines such as **GenPose++**, our method intentionally leverages *SE(3)-consistent* semantic features from large pretrained vision models (e.g., **DINO**, **RADIO**). These models have been extensively shown to provide robust, viewpoint- and lighting-invariant semantics, which serve as an effective and practical source of geometric consistency.
>
> Rather than assuming strict equivariance, our approach **inherits empirically validated SE(3)-robustness from the pretrained backbone**, incorporating this as a strong prior for pose estimation. This strategy avoids the added complexity and computational overhead of explicit equivariant modules, while still offering strong generalization under challenging conditions. Our ablation results in **Table 5** further demonstrate the effectiveness of these pretrained semantic features in improving robustness.
>
> ---
>
> **Weakness 2: The proposed MoE replaces the FFN and is argued to “specialize across diverse object types,” but there is no clear methodological support for this claim.**
>
> Thank you for the suggestion. Although the MoE (Mixture of Experts) module is not the principal contribution of this work, we conducted additional analyses to better understand its behavior. We examined expert activations across all layers and measured the selection frequency of each expert for different object categories.
>
> The results reveal several noticeable specialization patterns:
>
> - **Geometrically similar categories** (e.g., *bottle/bowl*, *mooncake/pitaya*) consistently route to similar expert subsets.
> - **Categories sharing coarse structural traits** (e.g., *laptop/calculator*, *toy_train/toy_boat*) exhibit partial but meaningful overlap in expert usage.
> - **Geometrically distinct categories** (e.g., *backpack vs. plug*, *knife vs. tomato*) display clearly divergent activation profiles.
>
> Heatmaps based on the cosine similarity of cross-category expert-usage vectors are provided in the appendix. These results suggest that, even with a standard MoE design, the experts tend to develop **geometry-aligned specialization**. This observation provides supporting evidence that the MoE may help improve generalization across diverse object types. We have included the corresponding analysis in **Appendix 7.7** of the revised paper.
>
> ---
>
> **Weakness 3：The multi-task objective simply sums all losses with equal weight for pose, size, shape, which raises the risk that one head dominates optimization.**
>
> Thank you for the suggestion. As stated in L276–277, *“In practice, we found that simply setting all loss coefficients to 1 yields stable training and strong performance, without requiring additional balancing.”*
>
> We also experimented with adjusting the loss weights and found that these changes did not produce any noticeable improvement in model performance. Therefore, we choose to keep all loss coefficients set to **1**, which provides both simplicity and stable training behavior.

---

### Official Review · Reviewer_9TJZ · 2025-10-31

**Soundness:** 4
**Presentation:** 4
**Contribution:** 3
**Rating:** 10
**Confidence:** 4

**Summary:**

This paper proposes a unified, category-agnostic framework for estimating an object’s 6D pose, size, and dense shape from a single RGB-D image. Unlike prior works that rely on object-specific templates or CAD models, the method performs reference-free inference by fusing dense 2D features from the RADIOv2.5 with partial 3D point clouds. The fused representation is processed by a Transformer encoder enhanced with Mixture-of-Experts (MoE) layers and decoded through two parallel heads for pose–size regression and shape reconstruction. The model achieves state-of-the-art results on seen categories, demonstrates strong zero-shot generalization to unseen objects, and operates in real time. The paper also introduces ObjaversePose, a new photorealistic dataset for open-set 6D understanding.

**Strengths:**

* The paper demonstrates a carefully reasoned combination of complementary modeling paradigms: foundation-model visual features (RADIOv2.5) for semantic generalization, DGCNN-based local geometric encoding for structure preservation, and Transformer-based global reasoning enhanced by a Mixture-of-Experts (MoE) mechanism for scalable specialization. The design is methodologically sound and practically effective
* The experimental validation is extensive and convincing. Evaluations span four benchmarks (SOPE, ROPE, ObjaversePose, HANDAL) covering both synthetic and real domains, as well as seen and unseen categories. The model consistently outperforms both category-level and reference-based baselines, even under severe occlusion and cross-domain shifts.
* The model achieves 28 FPS on commodity GPUs, a significant improvement over diffusion-based or multi-stage approaches such as GenPose++, which benefits the downstream tasks like robotic manipulation and embodied AI.
* The introduction of the ObjaversePose dataset substantially enriches the evaluation landscape for category-agnostic 6D perception.

**Weaknesses:**

* Although cross-domain generalization results are strong, the method remains trained entirely on synthetic data. It is unclear how performance scales to visually diverse, texture-rich, or long-tail real-world categories not represented in the synthetic domain.
* While success cases are well illustrated, the paper provides little insight into failure patterns (e.g., reflective surfaces, severe occlusion, or category ambiguity).

**Questions:**

* Quantitative evaluation under controlled photometric or geometric perturbations (e.g., sensor noise, lighting variation) would strengthen claims about deployment robustness.
* The RADIOv2.5 backbone is frozen during training. Have the authors investigated partial fine-tuning (e.g., LoRA or adapter layers) to improve transfer to 6D pose estimation task

---

> ### Author Response · Authors · 2025-11-26
>
> Dear reviewer 9TJZ, thank you for your review and valuable suggestions regarding our work. Please find our responses to your concerns below.
>
> ---
>
> **Question 1. Quantitative evaluation under controlled photometric or geometric perturbations (e.g., sensor noise, lighting variation) would strengthen claims about deployment robustness.**
>
> Thank you for suggestion.  To provide quantitative evidence of robustness under controlled perturbations, we introduce controlled perturbations to evaluate robustness across sensing modalities. For **depth**, we add heteroscedastic noise σ(z) = α + βz² with α ∈ {0.001, 0.002, 0.003} and β ∈ {0.000, 0.003, 0.006}, capturing the distance-dependent noise profiles of real sensors. For **RGB**, we apply exposure shifts (±0.5/±1.3/±2.2 EV), white-balance changes (±800/±2000/±4000 K), and illumination variations (gradient/vignetting at 0.10/0.25/0.40) to simulate common real-world lighting conditions.
>
> **Photometric Robustness (RGB)**
>
> Performance remained largely unaffected across the tested photometric settings, showing only a slight drop in accuracy. This indicates that the method is highly insensitive to lighting variations and remains stable under changes in exposure, color temperature, and illumination. We attribute this stability to the strong pretraining of the model’s backbone.
>
> **Geometric Robustness (Depth)**
>
> Depth noise leads to more noticeable but gradual performance degradation. The model remains robust under mild and moderate noise and maintains reliable accuracy even with severe heteroscedastic noise. This indicates that although depth perturbations affect performance more than photometric ones, the model still exhibits strong overall robustness.
>
> Overall, these controlled evaluations demonstrate that the method is **highly robust to photometric variations** and **maintains stable performance under realistic depth noise**, supporting the claimed deployment robustness. We have included this in **Appendix Section 7.5** of the revised paper.
>
> **Table 1 — Controlled robustness under photometric perturbations on RGB**
>
> | Perturbation | IoU25 | IoU50 | IoU75 | 5°2cm | 5°5cm | 10°2cm | 10°5cm | mean ΔAUC / ΔVUS (%) |
> | --- | --- | --- | --- | --- | --- | --- | --- | --- |
> | Clean (s=0) | 44.9 | 26.1 | 4.9 | 10.1 | 14.1 | 20.0 | 29.1 | +0.0 / +0.0 |
> | Light (low) | 44.7 | 26.0 | 4.8 | 10.1 | 14.1 | 19.9 | 28.8 | -0.5 / -0.6 |
> | Light (Moderate) | 44.5 | 25.7 | 4.7 | 10.0 | 13.9 | 19.7 | 28.5 | -1.3 / -1.6 |
> | Light (heavy) | 43.1 | 24.7 | 4.5 | 9.4 | 13.4 | 18.7 | 27.1 | -4.7 / -6.4 |
>
> **Table 2 — Controlled robustness under heteroscedastic depth noise**
>
> | Perturbation | α | β | IoU25 | IoU50 | IoU75 | 5°2cm | 5°5cm | 10°2cm | 10°5cm | mean ΔAUC / ΔVUS (%) |
> | --- | --- | --- | --- | --- | --- | --- | --- | --- | --- | --- |
> | Clean (s=0) | 0.000 | 0.000 | 44.9 | 26.1 | 4.9 | 10.1 | 14.1 | 20.0 | 29.1 | +0.0 / +0.0 |
> | Depth noise (low) | 0.001 | 0.000 | 44.7 | 26.0 | 4.8 | 10.0 | 14.1 | 20.1 | 29.2 | -0.5 / +0.1 |
> | Depth noise (Moderate) | 0.002 | 0.003 | 41.7 | 22.7 | 3.6 | 8.9 | 13.1 | 18.5 | 27.5 | -10.4 / -7.2 |
> | Depth noise (heavy) | 0.003 | 0.006 | 37.4 | 18.3 | 2.4 | 7.3 | 11.4 | 15.8 | 24.9 | -23.4 / -20.0 |
>
> ---
>
> **Weakness 2. While success cases are well illustrated, the paper provides little insight into failure patterns (e.g., reflective surfaces, severe occlusion, or category ambiguity).**
>
> Thank you for the valuable suggestion. In the main paper (Fig. 4), we show qualitative results demonstrating that our method remains robust even when point clouds are incomplete due to transparent or reflective objects. We further analyze three primary failure modes on ROPE: reflective materials, severe occlusion, and category ambiguity.
>
> **Reflective materials.**
>
> Our method exhibits minimal degradation on reflective objects, likely because the pretrained **RADIO** backbone provides features that compensate for missing geometric cues. This is consistent with the depth-only vs. RGB-D performance gap reported in **Table 5**. Failures mainly arise on *highly reflective* objects (e.g., kitchen knives) and *slender metallic objects* (e.g., pens).
>
> **Occlusion.**
>
> The model maintains strong performance under mild and moderate occlusion, as further illustrated in the supplementary video. However, when essential structural cues are missing, object orientation becomes ambiguous and the completeness of the reconstruction degrades.
>
> **Category ambiguity.**
>
> Most category-ambiguity failures fall into two scenarios:
>
> 1. **Highly symmetric or geometrically regular objects** (e.g., books, flutes), for which orientation is inherently difficult to disambiguate.
> 2. **Small objects** with low RGB resolution and limited geometric detail.
>
> These analyses provide clearer insight into the model's limitations. Additional examples and discussions have been included in **Appendix Section 7.6** of the revised paper.

---

> ### Author Response · Authors · 2025-11-26
>
> **Weakness 1. Although cross-domain generalization results are strong, the method remains trained entirely on synthetic data. It is unclear how performance scales to visually diverse, texture-rich, or long-tail real-world categories not represented in the synthetic domain.**
>
> Thank you for the suggestion. As noted in the **Future Work** section (L476–478), the performance of our model is inherently constrained by the range of categories included in the training data, and it may exhibit reduced accuracy when encountering *long-tail* or *atypical* shapes that are insufficiently represented in synthetic datasets. We acknowledge that robust generalization to long-tail categories remains an open and widely recognized challenge in open-set 6D pose estimation.
>
> To further assess this limitation, we evaluated our model on the representative subset of the **HomebrewDB** [1] validation set, one of the official datasets in the BOP Challenge, which contains more long-tail objects (e.g., minion toys, desk telephones, low-texture industrial parts) and exhibits greater variation in occlusion and clutter. As expected, performance decreases due to the difficulty of recognizing irregular or low-texture geometries. Nevertheless, the model still produces **semantically meaningful predictions** in most cases—such as inferring a toy’s orientation from the direction of its head—and continues to **outperform SOTA competing methods** under these challenging conditions.
>
> These observations confirm that long-tail generalization remains an open research problem. In the short term, **targeted data collection** and **task-specific fine-tuning** can mitigate this issue, while more fundamental advances will be required for long-term solutions.
>
> **Table 3 — Cross-Domain Generalization on Long-Tail Real-World Datasets**
>
> | Dataset | Method | IoU25 | IoU50 |
> | --- | --- | --- | --- |
> | HANDAL | GenPose++ | 16.7 | 4.3 |
> | HANDAL | **Ours** | **33.0** | **10.6** |
> | HomebrewDB | GenPose++ | 7.5 | 1.5 |
> | HomebrewDB | **Ours** | **17.4** | **5.2** |
>
> [1]  Kaskman, Roman, et al. "HomebrewedDB: RGB-D Dataset for 6D Pose Estimation of 3D Objects." arXiv preprint arXiv:1904.03167 (2019).
>
> ---
>
> **Question2. Discuss whether partial fine-tuning of the RADIOv2.5 backbone (e.g., LoRA or adapters) was explored and how it affects performance on 6D tasks.**
>
> We experimented with partial fine-tuning strategies using **LoRA**. Unexpectedly, these approaches degraded performance—not only on categories seen during training but even more substantially on unseen categories. This aligns with prior observations that **low-rank updates can distort geometry-sensitive representations** acquired during large-scale pretraining, thereby harming generalization.
>
> Following the design choices used in previous SOTA works such as **GenPose++**, we therefore **freeze the vision encoder** and train only the point-cloud encoder. This strategy preserves performance, maintains generality, and remains efficient in both training time and memory usage.

---

### Author Response · Authors · 2025-12-03

Dear **AC**, **SAC**, and **PC Members**,

We would like to express our sincere gratitude for your tremendous efforts in overseeing and managing the review process.

We thank all the **Reviewers** for their detailed and insightful comments on our work. We appreciate the reviewers for acknowledging our strengths and contributions, as well as their many helpful suggestions and questions. During the rebuttal phase, we have made substantial efforts to address all concerns raised. Specifically:

---

**1. Additional Analyses and Diagnostic Evaluations**

* Added a **failure-mode analysis** covering *reflective surfaces*, *severe occlusion*, and *category ambiguity* (**Appendix Sec. 7.6**).
* Added **t-SNE visualizations** illustrating semantic and geometric distributional differences between *ObjaversePose/HANDAL* and *SOPE*, supporting their roles as *unseen-category benchmarks* (**Appendix Sec. 7.4**).
* Added **Mixture-of-Experts (MoE) diagnostic evaluations**, including *expert-activation patterns* and *expert-selection heatmaps*, demonstrating specialization across diverse object types (**Appendix Sec. 7.7**).

---

**2. Expanded Experimental Evaluation and Robustness Studies**

* Conducted additional experiments on **HouseCat6D**, showing that our model—trained purely on synthetic data—achieves strong real-world generalization, including on *challenging transparent and reflective objects* (**L462–L472**).
* Added quantitative **robustness evaluations** under controlled RGB and depth perturbations, including *heteroscedastic depth noise*, *exposure shifts*, *white-balance variations*, and *illumination changes* (**Appendix Sec. 7.5**).
* Performed extended **cross-domain generalization** experiments on the expanded **HomebrewDB** *long-tail real-world dataset*.

---

**3. Methodological Clarifications and Design Justifications**

* Clarified the rationale for selecting **shape-reconstruction baselines**, ensuring consistency with prior work (**L450–L460**).
* Discussed potential finetuning of the **Radio backbone**, concluding that keeping it *frozen* is computationally efficient and preserves generalization.
* Clarified the motivation and significance of constructing and releasing the **ObjaversePose** dataset (**L292–L296**).
* Clarified the distinction between our setting and **Any6D**, emphasizing that our *unified*, *real-time*, *prior-free* framework achieves superior estimation of **pose**, **size**, and **shape** with strong generalization.
* Explained how our method inherits **SE(3)-robustness** from the pretrained backbone, making explicit SE(3) constraints unnecessary in our current design.

---

**4. Additional Implementation Details and Reporting Improvements**

We further provided clarifications regarding:

* training exclusively on **SOPE** without finetuning on other datasets,
* the **FPS measurement** protocol,
* **loss-weight** configurations,
* baseline choices in **Tables 1 and 2**,
* clarification of *n* and *d* at their first occurrence (**L194**).

---

An updated version of the main paper and supplementary materials has been uploaded, and all major changes are **highlighted in blue**.

Once again, we would like to extend our deepest gratitude for your dedicated efforts.

Best regards,

**Authors of Submission 14049**

---

### Meta-Review · Area_Chair_wAEi · 2025-12-30

**Summary:**

The paper proposes a unified, category-agnostic framework for estimating an object’s 6D pose, size, and shape from a single RGB-D image.

The initial reviewer scores were 10, 4, 2, 4. Reviewers raised several key concerns, regarding:
1) the proposed MoE design: missing implementation details, limited empirical evidence supporting its claimed advantages, and only marginal performance improvements;
2) the claim of “remarkably strong zero-shot generalization” to unseen real-world objects was not sufficiently substantiated;
3) the lack of discussion with closely related work Any6D.

In the rebuttal, the authors partially addressed these concerns. However, the novelty of the proposed approach and the strength of the generalization claims remain limited, particularly regarding absolute 6D pose estimation for unseen objects.

Therefore, the AC recommends rejecting the paper.

**Reviewer Concerns:**

See above.

**Reviewer Scores:**

See above.

---

### Decision · Program_Chairs · 2026-01-26

Reject